# Decision Stacks: Flexible Reinforcement Learning via Modular Generative Models

**Siyan Zhao**
Department of Computer Science
University of California Los Angeles
siyanz@cs.ucla.edu

**Aditya Grover**
Department of Computer Science
University of California Los Angeles
adityag@cs.ucla.edu

## Abstract

Reinforcement learning presents an attractive paradigm to reason about several distinct aspects of sequential decision making, such as specifying complex goals, planning future observations and actions, and critiquing their utilities. However, the combined integration of these capabilities poses competing algorithmic challenges in retaining maximal expressivity while allowing for flexibility in modeling choices for efficient learning and inference. We present Decision Stacks, a generative framework that decomposes goal-conditioned policy agents into 3 generative modules. These modules simulate the temporal evolution of observations, rewards, and actions via independent generative models that can be learned in parallel via teacher forcing. Our framework guarantees both expressivity and flexibility in designing individual modules to account for key factors such as architectural bias, optimization objective and dynamics, transferrability across domains, and inference speed. Our empirical results demonstrate the effectiveness of Decision Stacks for offline policy optimization for several MDP and POMDP environments, outperforming existing methods and enabling flexible generative decision making.[1]

## 1 Introduction

Modularity is a critical design principle for both software systems and artificial intelligence (AI). It allows for the creation of flexible and maintainable systems by breaking them down into smaller, independent components that can be easily composed and adapted to different contexts. For modern deep learning systems, modules are often defined with respect to their input and output modalities and their task functionalities. For example, Visual ChatGPT [Wu et al., 2023] defines a family of 22+ vision and language foundation models, such as ChatGPT [OpenAI, 2022] (language generation), SAM [Kirillov et al., 2023] (image segmentation), and StableDiffusion [Rombach et al., 2022] (text-to-image generation) for holistic reasoning over text and images. In addition to enabling new compositional applications, modularity offers the promise of interpretability, reusability, and debugging for complex workflows, each of which poses a major challenge for real-world AI deployments.

This paper presents progress towards scalable and flexible reinforcement learning (RL) through the introduction of a new modular probabilistic framework based on deep generative models. Prior work in modular RL focuses on spatiotemporal abstractions that simplify complex goals via hierarchical RL, e.g., [McGovern and Barto, 2001, Andreas et al., 2017, Simpkins and Isbell, 2019, Ahn et al., 2022, Kulkarni et al., 2016, Mendez et al., 2021]. Distinct but complementary to the prior lines of work, our motivating notion of modularity is based on enforcing token-level hierarchies in generative models of trajectories. In the context of RL, trajectories typically consist of a multitude of different tokens of information: goals, observations, rewards, actions. As shown in many recent works [Chen et al., 2021, Janner et al., 2021, 2022, Ajay et al., 2022, Zheng et al., 2022, Reed et al., 2022], we can effectively reduce RL to probabilistic inference [Levine, 2018] via learning deep generative models

---

[1]The project website and code can be found here: https://siyan-zhao.github.io/decision-stacks/

over token sequences. However, these frameworks lack any modular hierarchies over the different tokens leading to adhoc choices of generative architectures and objectives, as well as conditional independence assumptions that can be suboptimal for modeling long trajectory sequences.

We introduce Decision Stacks, a family of generative algorithms for goal-conditioned RL featuring a novel modular design. In Decision Stacks, we parameterize a distinct generative model-based module for future observation prediction, reward estimation, and action generation and chain the outputs of each module autoregressively. See Figure 1 for an illustration. While our factorization breaks the canonical time-induced causal ordering of tokens, we emphasize that the relative differences in different token types is significant to necessitate token-level modularity for learning effective policies and planners. Besides semantic differences, the different token types also show structural differences with respect to dimensionalities, domains types (discrete or continuous), modalities (e.g., visual observations, numeric rewards), and information density (e.g., rewards can be sparse, state sequences show relatively high continuity). Instead of modeling the token sequence temporally, parameterizing a distinct module for each token type can better respect these structural differences. In contrast, previous works like Decision Transformer Chen et al. [2021], Trajectory Transformer Janner et al. [2021] and Diffuser Janner et al. [2022] chained states, actions, and, in some cases, rewards within a single temporal model.

In practice, we can train each module in Decision Stacks independently using teacher forcing [Williams and Zipser, 1989], which avoids additional training time as the three modules can be trained in parallel. Decision Stacks shares similarities with many recent works [Janner et al., 2021, Ajay et al., 2022, Janner et al., 2022] that aim to reduce planning to sampling from a generative model. However, our modular design offers additional flexibility and expressivity. Each generative module itself is not restricted to being an autoregressive model and we experiment with modules based on transformers, diffusion models, and novel hybrids. Each generative modeling family makes tradeoffs in architecture, sampling efficiency, and can show varied efficacy for different data modalities. Decision Stacks also brings compositional generalization in scenarios where the modules can be reused across varied tasks or environments. A modular design that easily allows for the use of arbitrary generative models, along with an autoregressive chaining across the modules permits both flexibility and expressivity.

Empirically, we evaluate Decision Stacks on a range of domains in goal-conditioned planning and offline RL benchmarks for both MDPs and POMDPs. We find that the joint effect of modular expressivity and flexible parameterization in our models provides significant improvements over existing offline RL methods. This holds especially in partially observable settings, where Decision Stacks achieves a $15.7\%$ performance improvement over the closest baseline, averaged over 9 offline RL setups. We also demonstrate the flexibility of our framework by extensive ablation studies over the choice of generative architectures and inputs for each module.

## 2 Preliminaries

### 2.1 Goal conditioned POMDPs

We operate in the formalism of goal-conditioned Partially Observable Markov Decision Processes (POMDP) defined by the tuple $\mathcal{M} := (\mathcal{O}, \mathcal{S}, \mathcal{A}, \mathcal{G}, \mathcal{P}, \mathcal{R}, \mathcal{E}, \gamma, p_0(s), T)$. Respectively, $\mathcal{O}$ and $\mathcal{S}$ denote the observation space and the underlying state space, which are fully observable in the case of MDPs. $\mathcal{A}$, the action space, is consistent with that of MDPs. In the goal-condition setting, $\mathcal{G}$ specifies the task goal distribution which could be e.g., a language instruction or a visual destination state for multi-task policies, or a designed cumulative return for single-task policies. The transition probability function, $\mathcal{P} : \mathcal{S} \times \mathcal{A} \times \mathcal{S} \rightarrow [0, 1]$, describes the transition dynamics. Meanwhile, $\mathcal{R} : \mathcal{G} \times \mathcal{S} \times \mathcal{A} \mapsto \mathbb{R}$ defines the rewards that the decision-maker receives after performing action $a$ in state $s$. The observation emission model $\mathcal{E} = P(o|s)$, determines the probability of observing $o$ in state $s$. The $\gamma$, $p_0(s_0)$, and $T$ denote the discount factor [Puterman, 2014], initial latent state distribution, and horizon of an episode. In a POMDP context, the observations generated from the underlying state are intrinsically non-Markovian. The goal-conditioned RL objective is to find the optimal policy $\pi^*$ that maximizes the expected cumulative discounted reward over the episode horizon: $\eta_{\mathcal{M}} := \mathbb{E}_{G \sim \mathcal{G}, a_t \sim \pi(\cdot|s_t, g), s_{t+1} \sim \mathcal{P}(\cdot|s_t, a_t),} \left[ \sum_{t=0}^{T} \gamma^t r(s_t, a_t, G) \right].$

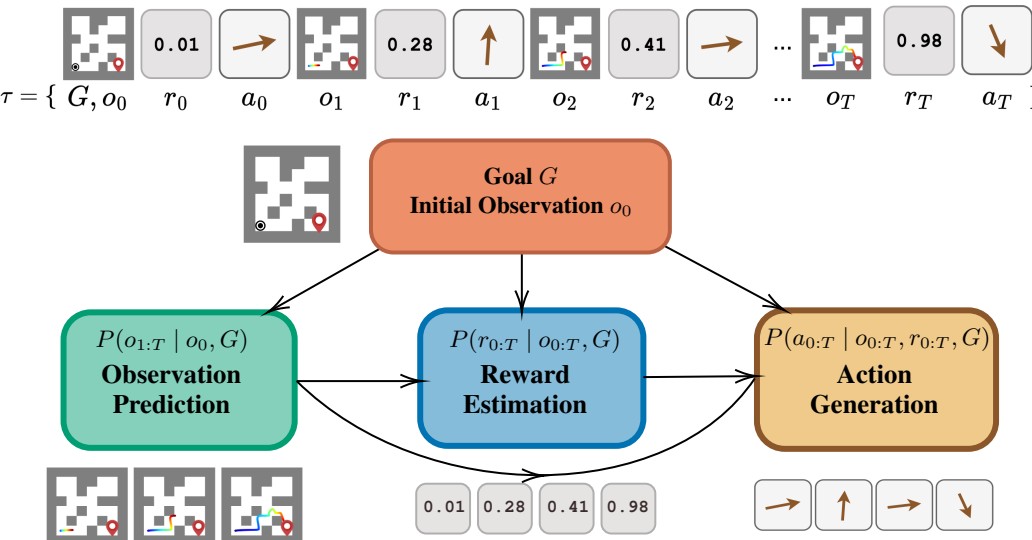

Figure 1: Illustration for the Decision Stacks framework for learning reinforcement learning agents using probabilistic inference. In contrast to a time-induced ordering, we propose a modular design that segregates the modeling of observation, rewards, and action sequences. Each module can be flexibly parameterized via any generative model and the modules are chained via an autoregressive dependency graph to provide high overall expressivity.

## 2.2 Offline reinforcement learning

Offline reinforcement learning (RL) is a paradigm for policy optimization where the agent is only given access to a fixed dataset of trajectories and cannot interact with the environment to gather additional samples. Offline RL can be useful in domains where collecting data online is challenging or infeasible such as healthcare [Murphy et al., 2001] and autonomous driving. A major obstacle in offline RL is dealing with distributional shifts. If we naively use Bellman backup for learning the Q-function of a given policy, the update which relies on the actions sampled from policy $\pi$ can learn inaccurately high values for out-of-distribution actions, leading to instability in the bootstrapping process and causing value overestimation [Kumar et al., 2020].

## 2.3 Generative models: Autoregressive transformers and Diffusion models

In this work, we are interested in learning the distribution $p_{\text{data}}(\mathbf{x}|\mathbf{c})$ using a dataset $D$ consisting of trajectory samples $\mathbf{x}$ and conditioning $\mathbf{c}$. We consider two conditional generative models for parameterizing our agent policies to learn the distribution:

**Transformer** is a powerful neural net architecture for modeling sequences [Vaswani et al., 2017]. It consists of multiple identical blocks of multi-head self-attention modules and position-wise fully-connected networks. The vanilla transformer can be modified with a causal self-attention mask to parameterize an autoregressive generative model as in GPT [Radford et al., 2018]. Autoregressive generative models, such as the transformers, factorize the joint distribution $p(x_1, \ldots, x_n)$ as a product of conditionals, which can be represented as: $p(\mathbf{x}) = \prod_{i=1}^{n} p(x_i|\mathbf{x}_{<i})$. This equation shows that the probability of each variable $x_i$ depends on the previous variables $x_1, \ldots, x_{i-1}$. One advantage of this factorization is that each conditional probability can be trained independently in parallel via teacher forcing [Williams and Zipser, 1989]. In autoregressive generation, sampling is done sequentially, where each variable $x_i$ is sampled based on its preceding variables.

**Diffusion Models** [Sohl-Dickstein et al., 2015, Ho et al., 2020] are latent variable models that consist of a predefined forward noising process $q(\boldsymbol{x}_{k+1}|\boldsymbol{x}_k) := \mathcal{N}(\boldsymbol{x}_{k+1}; \sqrt{\alpha_k}\boldsymbol{x}_k, (1-\alpha_k)I)$ that gradually corrupts the data distribution $q(\boldsymbol{x}_0)$ into $\mathcal{N}(0, I)$ in $K$ steps, and a learnable reverse denoising process $p_\theta(\boldsymbol{x}_{k-1}|\boldsymbol{x}_k) := \mathcal{N}(\boldsymbol{x}_{k-1}|\mu_\theta(\boldsymbol{x}_k, k), \Sigma_k)$. For sampling, we first generate a latent sample from the Gaussian prior $\mathcal{N}(0, I)$ and gradually denoise it using the learned model $p_\theta(\boldsymbol{x}_{k-1}|\boldsymbol{x}_k)$ for

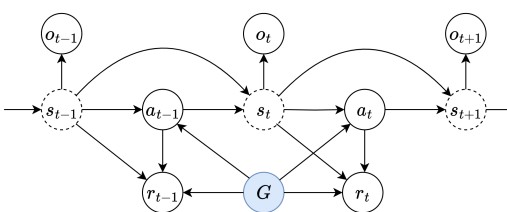

Figure 2: Graphical model for the data generation process in a POMDP. Here, we show the case where the behavioral policy can potentially act based on hidden state information. The dashed circles implies that this state information is not stored in the offline dataset. $G$ represents the task conditioning, e.g., a target return (for single-task agents) or a navigation goal (for multi-task agents).

$K$ steps to obtain the data sample $\boldsymbol{x}_0$. Diffusion models can be extended for conditional generation using *classifier-free guidance* [Ho and Salimans, 2022] where a conditional $\epsilon_\theta\left(\boldsymbol{x}_k, \mathbf{c}, k\right)$ and an unconditional $\epsilon_\theta\left(\boldsymbol{x}_k, k\right)$ noise model is trained. Conditional data is sampled with the perturbed noise $\epsilon_\theta\left(\boldsymbol{x}_k, k\right) + \omega\left(\epsilon_\theta\left(\boldsymbol{x}_k, \mathbf{c}, k\right) - \epsilon_\theta\left(\boldsymbol{x}_k, k\right)\right)$, where $\omega$ is the guidance strength and $\mathbf{c}$ is the conditioning information.

## 3 Flexible and Modular RL via Decision Stacks

In Figure 2, we consider a directed graphical model for the data generation process in Partially Observable Markov Decision Processes (POMDP) Kaelbling et al. [1998]. The environment encodes the underlying state transitions $P(s_{t+1}|s_t, a_t)$, goal-dependent reward function $P(r_t|s_t, a_t, G)$, and the observation emission probability $P(o_t|s_t)$. Unlike the learning agent which only has access to observations, the behavioral policy used for generating the offline trajectory dataset might also have access to the hidden state information. Such a situation is common in real-world applications, e.g., a human demonstrater may have access to more information about its internal state than what is recorded in video demonstration datasets available for offline learning. Finally, note that a Markov Decision Process (MDP) can be viewed as a special case of a POMDP, where the observation at each timestep, $o_t$, matches the underlying state, $s_t$. To avoid ambiguity, we overload the use of $o_t$ to denote both the state and observation in an MDP at time $t$.

In the context of goal-conditioned decision-making, a finite-horizon trajectory in the offline dataset $\mathcal{D}$ is composed of a goal $G$ and a sequence of observations, actions, and reward tokens.

$$\tau = (G, o_0, a_0, r_0, \ldots, o_t, a_t, r_t, \ldots, o_T, a_T, r_T). \tag{1}$$

Our primary objective lies in learning a goal-conditioned distribution for $P_{\text{data}}(a_{0:T}, o_{1:T}, r_{0:T}|o_0, G)$ conditioned on an arbitrary goal $G \in \mathcal{G}$ and an initial observation $o_0 \in \mathcal{O}$. Leveraging the chain rule of probability, we can factorize this joint distribution into a product of conditional probabilities. For example, Janner et al. [2021] use a time-induced autoregressive factorization:

$$P_{\text{data}}\left(a_{0:T}, o_{1:T}, r_{0:T} \mid o_0, G\right) \approx \prod_{t=1}^{T} P_\theta(o_t|\tau_{<t}) \prod_{t=0}^{T} P_\theta(r_t|o_t, \tau_{<t}) \prod_{t=0}^{T} P_\theta(a_t|o_t, r_t, \tau_{<t}) \tag{2}$$

where $\tau_{<t}$ denotes all the tokens in the trajectory before time $t$. Each conditional factor is parameterized via an autoregressive transformer with shared parameters $\theta$. If the parameterization is sufficiently expressive, any choice of ordering for the variables suffices. However, in practice, we are limited by the size of our offline dataset and the choice of factorization can play a critical role.

In Decision Stacks, we propose to use a modular factorization given as:

$$P_{\text{data}}\left(a_{0:T}, o_{1:T}, r_{0:T} \mid o_0, G\right) \approx \underbrace{P_{\theta_1}\left(o_{1:T} \mid o_0, G\right)}_{\text{observation module}} \cdot \underbrace{P_{\theta_2}\left(r_{0:T} \mid o_{0:T}, G\right)}_{\text{reward module}} \cdot \underbrace{P_{\theta_3}\left(a_{0:T} \mid o_{0:T}, r_{0:T}, G\right)}_{\text{action module}}. \tag{3}$$

Each of the 3 modules (observations, rewards, or actions) focuses on predicting a distinct component of the POMDP and has its own set of parameters $(\theta_1, \theta_2, \theta_3)$. Our motivation stems from the fact that in real-world domains, each component is sufficiently distinct from the others in its semantics and representation. Such variances span across a multitude of factors including dimensionalities, domain types (discrete or continuous), modalities (e.g., visual observations, numeric rewards), and information density (e.g., rewards can be sparse, state sequences show relatively high continuity).

**Modular Expressivity.** In Eq. 3, each module is chained autoregressively with the subsequent modules. This is evident as the output variables of one module are part of the input variables for all the subsequent modules. Under idealized conditions where we can match each module to the data conditional, this autoregressive structure enables maximal expressivity, as autoregressive models derived from the chain rule of probability can in principle learn any data distribution given enough model capacity and training data. Further, our explicit decision to avoid any parameter sharing across modules also permits trivial hardware parallelization and transfer to new environments with shared structure.

**Flexible Generative Parameterization.** Since each module predicts a sequence of objects, we use any deep generative model for expressive parameterization of each module. Our experiments primarily focus on autoregressive transformers and diffusion models. We also consider hybrid combinations, as they are easy to execute within our framework and can avoid scenarios where individual model families suffer, e.g., diffusion models lag behind transformer for discrete data; whereas transformers are generally poor for modeling continuous signals such as image observations. In real-world environments, many of these challenges could simultaneously occur such as agents executing discrete actions given continuous observations. Finally, each module is conditioned on a goal. For training a multi-task agent, the goal can be specified flexibly as spatial coordinates, a visual image, a language instruction, etc. For single-task agents, we specify the goal as the trajectory returns during training and desired expert-valued return during testing, following prior works in return-conditioned offline RL [Chen et al., 2021, Emmons et al., 2021].

**Learning and Inference.** Given an offline dataset, each module can be trained in parallel using teacher forcing [Williams and Zipser, 1989]. At test-time, our framework naturally induces a planner, as in order to predict an action at time $t$, we also need to predict the future observations and rewards. We can execute either an open-loop or closed-loop plan. Open-loop plans are computationally efficient as they predict all future observations and rewards at once, and execute the entire sequence of actions. In contrast, a closed-loop plan is likely to be more accurate as it updates the inputs to the modules based on the environment outputs at each time-step. Using a closed-loop plan, we can sample the action at time $t$ as follows:

$$\hat{o}_{t+1:T} \sim P_{\theta_1}\left(o_{t+1:T} \mid o_{0:t}, G\right) \tag{4}$$

$$\hat{r}_{t+1:T} \sim P_{\theta_2}\left(r_{t+1:T} \mid r_{0:t}, o_{0:t}, \hat{o}_{t+1:T}, G\right) \tag{5}$$

$$\hat{a}_t \sim P_{\theta_3}\left(a_t \mid a_{0:t-1}, o_{0:t}, \hat{o}_{t+1:T}, r_{0:t}, \hat{r}_{t+1:T}, G\right) \tag{6}$$

The hat symbol (^) indicates predicted observations, rewards, and actions, while its absence denotes observations, rewards, and actions recorded from the environment and the agent in the previous past timesteps. For closed-loop planning, Eqs. 4, 5, 6 require us to condition the joint observation, reward and action distributions on the past trajectory tokens. For a module that is parameterized autoregressively, this is trivial as we can simply choose a time-induced ordering and multiply the conditionals for the current and future timesteps. For example, if the observation module is an autoregressive transfer, then we can obtain the sampling distribution in Eq. 4 as: $P_{\theta_1}\left(o_{t+1:T} \mid o_{0:t}, G\right) = \prod_{i=t+1}^{T} P_{\theta_1}\left(o_i \mid o_{<i}, G\right)$. For a diffusion model, this task is equivalent to inpainting and can be done by fixing the environment observations until time $t$ at each step of the denoising process [Janner et al., 2022].

**Distinction with Key Prior Works.** We will include a more detailed discussion of broad prior works in §5 but discuss and contrast some key baselines here. While the use of generative models for goal-conditioned offline RL is not new, there are key differences between Decision Stacks and recent prior works. First, we choose a planning approach unlike other model-free works, such as Decision Transformers [Chen et al., 2021] and diffusion-based extensions [Wang et al., 2022]. Second, there exist model-based approaches but make different design choices; Trajectory Transformer [Janner et al., 2021] uses a time-induced causal factorization parameterized by a single autoregressive transformer, Diffuser [Janner et al., 2022] uses diffusion models over stacked state and action pairs, Decision Diffuser [Ajay et al., 2022] uses diffusion models for future state prediction and an MLP-based inverse dynamics model to extract actions. Unlike these works, we propose a modular structure that is maximally expressive as it additionally models reward information and does not make any conditional independence assumption for the state, reward and action modules. As our experiments demonstrate, the modular expressivity and architectural flexibility in Decision Stacks are especially critical for goal-conditioned planning and dealing with partial observability.

Table 1: Performance on Maze2D tasks. DS significantly outperforms other baselines without the need for a handcoded controller. Note that DD and DS share the same diffusion-based observation model architecture and hence with a handcoded controller, their performance is the same. We average the results over 15 random seeds and emphasize in bold scores within 5 percent of the maximum per task ($\geq 0.95 \cdot \max$).

| Task | Environment | MPPI | CQL | IQL | Diffuser | DD | DS | Diffuser with Handcoded Controller | DS / DD with Handcoded Controller |
|---|---|---|---|---|---|---|---|---|---|
| Single Goal | umaze | 33.2 | 5.7 | 47.4 | 86.9 ±26.4 | **113.8** ±11.3 | 111.3 ±12.2 | 113.9 ±3.1 | 119.5 ±2.6 |
| | medium | 10.2 | 5.0 | 34.9 | 108.5 ±17.4 | 103.7 ±21.2 | 111.7 ±2.4 | 121.5 ±2.7 | 112.9 ±11.8 |
| | large | 5.1 | 12.5 | 58.6 | 45.4 ±14.5 | 111.8 ±43.4 | **171.6** ±13.4 | 123.0 ±6.4 | 132.8 ±21.0 |
| | **Average** | 16.2 | 7.7 | 47.0 | 80.2 | 109.8 | **131.5** | 119.5 | 121.7 |
| Multi Goals | umaze | 41.2 | - | 24.8 | 114.4 ±16.3 | 105.6 ±14.5 | 121.3 ±12.2 | 129.0 ±1.8 | **136.1** ±4.2 |
| | medium | 15.4 | - | 12.1 | 54.6 ±14.5 | **126.4** ±14.3 | 122.3 ±3.7 | 127.2 ±3.4 | 124.6 ±11.3 |
| | large | 8.0 | - | 13.9 | 41.0±20.1 | 116.0 ±33.1 | 126.7 ±21.8 | 132.1 ±5.8 | **134.8** ±12.3 |
| | **Average** | 21.5 | - | 16.9 | 70.0 | 111.6 | 123.4 | 129.4 | 131.8 |

# 4 Experiments

Our experiments aim to answer the following questions:
§4.1 How does Decision Stacks perform for long-horizon multi-task planning problems?
§4.2 How does Decision Stacks compare with other offline RL methods in MDP environments?
§4.3 How does Decision Stacks compare with other offline RL methods in POMDP environments?
§4.4 How does the architectural feasibility for each module affect downstream performance? How does the modularity enable compositional generalization? How important is the role of reward modeling for Decision Stacks?

For §4.1, §4.2, and §4.3, we experiment with D4RL environments and parameterize Decision Stacks with a diffusion-based observation model, an autoregressive transformer-based reward model, and an autoregressive transformer-based action model. Finally, in §4.4, we will ablate the full spectrum of architecture design choices for each module.

## 4.1 Long-Horizon Goal-Conditioned Environments

We first test for the planning capabilities of Decision Stacks on the Maze2D task from the D4RL [Fu et al., 2020] benchmark. This is a challenging environment requiring an agent to generate a plan from a start location to a goal location. The demonstrations contain a sparse reward signal of +1 only when the agent reaches close to the goal. Following Janner et al. [2022], we consider 2 settings. In the Single Goal setting, the goal coordinates are fixed, and in the Multi Goal setting, the goals are randomized at test-time. We compare against classic trajectory optimization techniques that have knowledge of the environment dynamics (MPPI [Williams et al., 2015], extensions of model-free RL baselines (CQL [Kumar et al., 2020] and IQL [Kostrikov et al., 2021]), and the two most closely related works in generative planning based on diffusion models: Diffuser [Janner et al., 2022] and Decision Diffuser (DD) [Ajay et al., 2022].

We show our results in Table 1 for different goal types and maze grids. While Janner et al. [2022] previously demonstrated remarkable ability in generating long-horizon plans using Diffuser, their trajectory plans were executed by a handcoded controller. However, we experimentally found that Diffuser and DD's own generated actions fail to perfectly align with their generated plans, as shown in the example rollouts in Figure 3. We hypothesize this could stem from the lack of modularity in Diffuser affecting the generation fidelity, or the lack of expressivity in using an MLP-based inverse dynamics model in DD which limits the context length required for long-horizon planning. In contrast, we find that DS generates robust trajectory plans and matching action sequences with significant improvements over baselines.

## 4.2 Offline Reinforcement Learning Performance in MDPs

Next, we examine the performance of Decision Stacks in offline RL tasks across various high-dimensional locomotion environments from the D4RL offline benchmark suite [Fu et al., 2020] in Table 2. We compare Decision Stacks (DS) with other offline RL algorithms including imitation learning via Behavior Cloning (BC), value-based approaches like IQL [Kostrikov et al., 2021] and CQL [Kumar et al., 2020], model-based algorithm MOReL [Kidambi et al., 2020], transformer-based

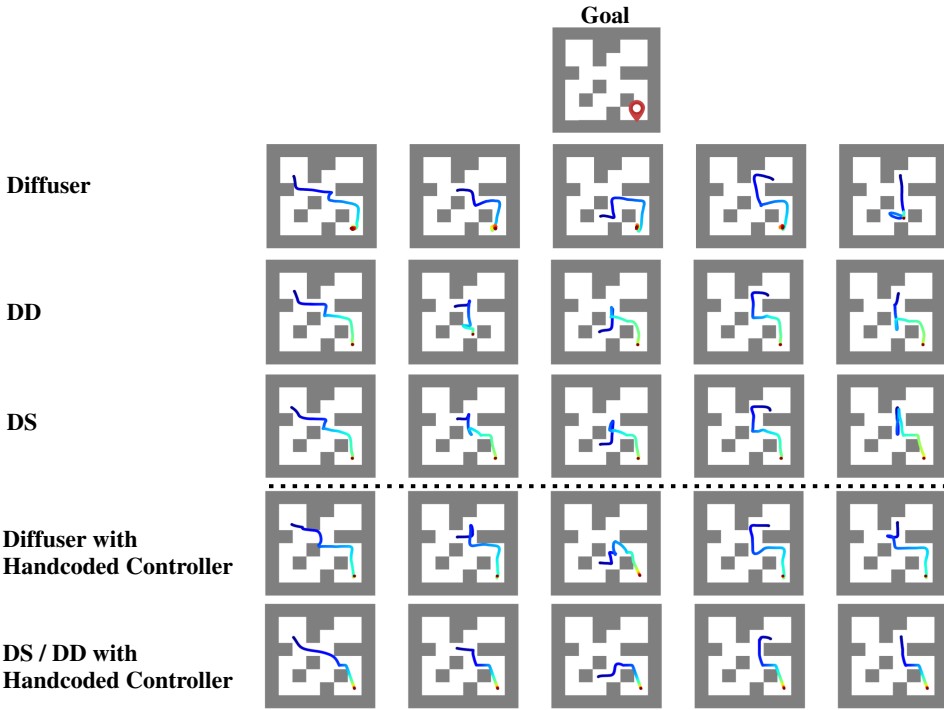

Figure 3: Example rollouts on the Maze2D-medium-v1 environment. The goal is located at the bottom right corner of the maze. The trajectory waypoints are color-coded, transitioning from blue to red as time advances. The bottom two rows demonstrates that Diffuser, DD, and DS are all capable of generating good plans that can be executed well with a handcoded controller. However, the respective action models result in differing executions. Compared to DD and Diffuser, DS generates smoother trajectories that are more closely aligned with the future waypoints planned by the observation model.

generative models such as Decision Transformer (DT) [Chen et al., 2021] and Trajectory Transformer (TT) [Janner et al., 2021], and diffusion-based generative models Diffuser [Janner et al., 2022] and Decision Diffuser (DD) [Ajay et al., 2022]. In our evaluation, we also included our reproduced scores for DD. DD uses the same architecture for observation prediction as Decision Stacks and is hence, the closest baseline. However, we found its performance to be sensitive to return conditioning and in spite of an extensive search for hyperparameters and communication with the authors, our reproduced numbers are slightly lower. We provide more details in the Appendix. For a fair comparison, we used the same set of hyperparameters that give the best performance for the DD baseline.

We show results averaged over 15 planning seeds and normalize the scores such that a value of 100 represents an expert policy, following standard convention [Fu et al., 2020]. Decision Stacks outperforms or is competitive with the other baselines on 6/9 environments and is among the highest in terms of aggregate scores. These results suggest that even in environments where we can make appropriate conditional independence assumptions using the MDP framework, the expressivity in the various modules of Decision Stacks is helpful for test-time generalization.

## 4.3 Offline Reinforcement Learning Performance in POMDPs

Next we consider the POMDP setting where the logged observations are incomplete representations of the underlying states. To generate the POMDPs datasets, we exclude the two velocity dimensions from the full state representation, which consists of both positions and velocities. This simulates a lack of relevant sensors. A sensitivity analysis on dimension occlusions further strengthens our results, as shown in Appendix Table 7. DS continues to outperform other baselines for each environment from the D4RL locomotion datasets. We report our results in Table 3 and compare against other generative baselines. Decision Stacks (DS), consistently achieves competitive or superior results compared to the other algorithms, including BC, DT, TT, and DD. Notably, DS outperforms other methods in

Table 2: **Offline Reinforcement Learning Performance in MDP**. Our results are averaged over 15 random seeds. Following Kostrikov et al. [2021], we bold all scores within 5 percent of the maximum per task ($\geq 0.95 \cdot \max$).

| Dataset | Environment | BC | IQL | CQL | DT | TT | MOReL | DD | DD (reproduced) | Diffuser | DS (ours) |
|---|---|---|---|---|---|---|---|---|---|---|---|
| Medium-Expert | HalfCheetah | 55.2 | 86.7 | **91.6** | 86.8 | **95.0** | 53.3 | 90.6 | **91.5** ±2.5 | 79.8 | **95.7** ±0.3 |
| Medium-Expert | Hopper | 52.5 | 91.5 | 105.4 | **107.6** | **110.0** | 108.7 | **111.8** | **111.6** ±2.8 | 107.2 | **107.0** ±3.2 |
| Medium-Expert | Walker2d | **107.5** | **109.6** | **108.8** | **108.1** | 101.9 | 95.6 | **108.8** | **105.2** ±2.3 | **108.4** | **108.0** ±0.1 |
| Medium | HalfCheetah | 42.6 | **47.4** | 44.0 | 42.6 | **46.9** | 42.1 | **49.1** | 46.4 ±5.1 | 44.2 | **47.8** ±0.4 |
| Medium | Hopper | 52.9 | 66.3 | 58.5 | 67.6 | 61.1 | **95.4** | 79.3 | 81.2 ±7.2 | 58.5 | 76.6 ±4.2 |
| Medium | Walker2d | 75.3 | 78.3 | 72.5 | 74.0 | 79.0 | 77.8 | **82.5** | **79.9** ±5.3 | 79.7 | **83.6** ±0.3 |
| Medium-Replay | HalfCheetah | 36.6 | **44.2** | **45.5** | 36.6 | 41.9 | 40.2 | 39.3 | 39.4 ±1.5 | 42.2 | 41.1 ±0.1 |
| Medium-Replay | Hopper | 18.1 | 94.7 | **95.0** | 82.7 | 91.5 | 93.6 | **100** | **95.3** ±3.7 | 96.8 | 89.5 ±4.2 |
| Medium-Replay | Walker2d | 26.0 | 73.9 | 77.2 | 66.6 | **82.6** | 49.8 | 75 | 72.3 ±3.1 | 61.2 | **80.7** ±1.5 |
| **Average** | | 51.9 | 77.0 | 77.6 | 74.7 | **78.9** | 72.9 | 82.2 | 80.3 | 75.3 | 81.1 |

most environments and attains the highest average score of 74.3, which reflects a 15.7% performance improvement over the next best-performing approach Diffuser. This highlights the effectiveness of our approach in handling POMDP tasks by more expressively modeling the dependencies among observations, actions, and rewards.

Table 3: **Offline Reinforcement Learning Performance in POMDP**. Our results are averaged over 15 random seeds. Following Kostrikov et al. [2021], we bold all scores within 5 percent of the maximum per task ($\geq 0.95 \cdot \max$).

| Dataset | Environment | BC | DT | TT | DD | Diffuser | DS (ours) |
|---|---|---|---|---|---|---|---|
| Medium-Expert | HalfCheetah | 42.1 | 80.8 | **94.9** | 19.07 | 82.2 | **92.7** ±0.8 |
| Medium-Expert | Hopper | 51.1 | 105.2 | 61.6 | 32.7 | 70.7 | **110.9** ±0.4 |
| Medium-Expert | Walker2d | 51.3 | **106.0** | 51.7 | 74.8 | 82.4 | 94.1 ±8.5 |
| Medium | HalfCheetah | 43.3 | 42.7 | **46.7** | 40.3 | **45.4** | **47.1** ±0.3 |
| Medium | Hopper | 36.4 | **63.1** | 55.7 | 38.1 | **62.2** | 57.7 ±3.9 |
| Medium | Walker2d | 39.4 | 64.2 | 28.5 | 53.2 | 55.7 | **74.3** ±4.2 |
| Medium-Replay | HalfCheetah | 2.1 | 35.5 | **43.8** | 39.8 | 39.3 | 40.3 ±1.2 |
| Medium-Replay | Hopper | 24.3 | 78.3 | **84.4** | 22.1 | 80.9 | **86.9** ±2.6 |
| Medium-Replay | Walker2d | 23.8 | 45.3 | 10.2 | 58.4 | 58.7 | **66.8** ±1.8 |
| **Average** | | 34.9 | 47.9 | 53.0 | 42.0 | 64.2 | **74.3** |

## 4.4 Architectural Flexibility and Compositional Generalization

Table 4: Performance on Hopper-medium-v2 POMDP using various reward and action models, with **diffusion-based** or **transformer-based** observation model. In each choice of observation model, the algorithm with the highest performance is highlighted.

| Reward models | Action models | | | | | |
|---|---|---|---|---|---|---|
| | Transformer | Diffusion | MLP | Transformer | Diffusion | MLP |
| Transformer | 57.7 ±3.9 | **58.2** ±4.3 | 45.6 ±4.1 | 53.0 ±3.7 | 54.3 ±3.3 | 36.7 ±4.2 |
| Diffusion | 51.7 ±1.7 | 56.9 ±2.2 | 36.3 ±3.1 | **58.0** ±4.4 | 46.9 ±3.7 | 34.9 ±3.5 |
| MLP | 56.0 ±3.5 | 52.6 ±2.5 | 33.3 ±3.0 | 55.0 ±3.9 | 52.1 ±2.7 | 42.5 ±4.1 |
| | Diffusion-based **observation model** | | | Transformer-based **observation model** | | |

Decision Stacks distinctly separates the prediction of observations, rewards, and actions employing three distinct models that can be trained independently using teacher forcing. In this section, we explore the additional flexibility offered by different architecture choices for each module. For observation, reward, and action prediction, we consider diffusion models and Transformer-based autoregressive models. For reward and action models, we additionally consider MLPs that are restricted in their window and only look at the immediate state information to make a decision. The

Table 5: Ablation results comparing the performance of action models with and without reward information as input, across different architectures, in the dense reward POMDP task of Hopper-medium-v2. The results suggests a clear advantage when incorporating reward modeling.

| Observation model | Action models without reward modelling | | | Action models with reward modelling | | |
|---|---|---|---|---|---|---|
| | Transformer | Diffusion | MLP | Transformer | Diffusion | MLP |
| Diffusion-based | 43.6 ±1.3 | 43.7 ±3.4 | 38.1 ±2.1 | 57.7 ±3.9 | 58.2 ±4.3 | 45.6 ±4.1 |
| Transformer-based | 45.1 ±5.2 | 39.4 ±3.2 | 39.6 ±3.7 | 58.0 ±4.4 | 54.3 ±3.3 | 42.5 ±4.1 |

results shown in Table 4 display a combination of 2x3x3 policy agents for the Hopper-medium v2 POMDP environment. Since we adopt a modular structure, we can compose the different modules efficiently and hence, we only needed to train 2 (state) + 3 (reward) + 3 (action) models. In Table 4, we find that the performance of pure transformer- or diffusion-based Decision Stacks gives reasonable performance (transformers: 53.0, diffusion: 56.9) but these pure combinations can be slightly outperformed by hybrids, e.g., the best achieving entry (58.2) in Table 4 uses a diffusion-based obsevation model, a transformer-based reward model and a diffusion-based action model. MLPs generally are outperformed by generative architectures, especially when used for modeling actions.

In addition to architectural flexibility, DS's modularity also supports compositional transfer across tasks. The lack of parameter sharing across modules not only enables hardware parallelization but also enables the transfer of modules to new environments. As shown in table 6, we created modified versions of the Maze2d-Umaze-v2 environment, altering action and reward spaces while keeping the observation model consistent. Both the reward and observation models were kept consistent during transfers between different action spaces. A single observation model was shared across six environments, and each of the two reward models was utilized across three action spaces. Our findings show that DS promotes efficient learning and compositional generalization, emphasizing its modular efficiency in scenarios with reusable components across varied tasks or environments.

Table 6: Experiments on Compositional Generalization. In the Maze2d-Umaze-v2 environment, we analyze both dense and sparse rewards. For the action space, we consider three variations of the force on a 2D ball: 1) Unscaled Force: the original force applied to the ball in the 2D maze environment; 2) Reflected Force: the original force mirrored across the x-axis; 3) Rotated Force: the original force rotated 30 degrees counterclockwise, altering the direction of the applied force. Each row shares the same reward model and observation model across the three different action spaces. The results underscore the modular efficiency of DS in scenarios where reusable components exist across different tasks or environments. The standard error is reported, and the results are averaged over 15 random seeds.

| | | Action Space | | |
|---|---|---|---|---|
| | | Unscaled Force | Reflected Force | Rotated Force |
| Reward Space | dense | $93.2 \pm 10.7$ | $90.3 \pm 9.1$ | $102.9 \pm 8.1$ |
| | sparse | $111.3 \pm 12.2$ | $102.6 \pm 3.9$ | $115.2 \pm 8.0$ |

Furthermore, we compare the best reward modeling architectures with alternatives that do not consider rewards. This is standard practice for Diffuser [Janner et al., 2022] and Decision Diffuser (DD) [Ajay et al., 2022]. For example, DD predict the action at time $t$ only based on the current observation and next observation, $P(a_t|o_t, o_{t+1})$, parameterized via an MLP. As delineated in Table 5, the inclusion of reward models significantly boosts performance in the dense reward POMDP environment. We include additional analysis and discussion in the Appendix D.

## 5 Related works

**Offline Reinforcement Learning** is a paradigm that for learning RL policies directly from previously logged interactions of a behavioral policy. The key challenge is that any surrogate models trained on an offline dataset do not generalize well outside the dataset. Various strategies have been proposed to mitigate the challenges due to distribution shifts by constraining the learned policy to be conservative

and closely aligned with the behavior policy. These include learning a value function that strictly serves as a lower bound for the true value function in CQL [Kumar et al., 2020], techniques focus on uncertainty estimation such as Kumar et al. [2019], and policy regularization methods [Wu et al., 2019, Ghasemipour et al., 2021, Kumar et al., 2019, Fujimoto and Gu, 2021, Fujimoto et al., 2019]. Model-based methods like MORel [Kidambi et al., 2020] and ReBeL [Lee et al., 2021] add pessimism into the dynamics models. In the context of partially observed settings, Rafailov et al. [2021] extends model-based offline RL algorithms by incorporating a latent-state dynamics model for high-dimensional visual observation spaces, effectively representing uncertainty in the latent space and Zheng et al. [2023] derive algorithms for offline RL in settings where trajectories might have missing actions. Our work takes the RL as inference perspective [Levine, 2018] and employs the tools of probabilistic reasoning and neural networks for training RL agents.

**Generative models for offline RL.** Over the past few years, the RL community has seen a growing interest in employing generative models for context-conditioned sequence generation by framing the decision-making problem as a generative sequence prediction problem. Here, we expand on our discussion from §3 with additional context and references. Decision Transformer [Chen et al., 2021] and Trajectory Transformer [Janner et al., 2021] concurrently proposed the use of autoregressive transformer-based models [Radford et al., 2018] for offline RL in model-based and model-free setups. Online decision transformer [Zheng et al., 2022] further finetunes the offline pretrained policies in online environments through a sequence-level exploration strategy. GATO [Reed et al., 2022, Lee et al., 2022] and PEDA [Zhu et al., 2023] scale these models to multi-task and multi-objective settings. MaskDP Liu et al. [2022] shows that other self-supervised objectives such as masking can also enable efficient offline RL, especially in goal-conditioned settings. These advancements have also been applied to other paradigms in sequential decision making such as black-box optimization and experimental design [Nguyen and Grover, 2022, Krishnamoorthy et al., 2023a,b]. Recent works have shown that changing the generative model from transformer to a diffuser with guidance can improve performance in certain environments and also permit planning for model-based extensions [Janner et al., 2021, Ajay et al., 2022, Wang et al., 2022, Chen et al., 2022]. Dai et al. [2023] considers an extension where the state model is a pretrained text2image model and actions are extracted from consecutive image frames. As discussed in §3, these works make specific design choices that do not guarantee the modularity, flexibility, and expressivity ensured by our framework.

# 6 Conclusion

We proposed Decision Stacks, a modular approach for learning goal-conditioned policies using offline datasets. Decision Stacks comprises of 3 modules tasked with the prediction of observations, rewards, and actions respectively. In doing so, we strive for the twin benefits of expressivity through autoregressive conditioning across the modules and flexibility in generative design within any individual module. We showed its empirical utility across a range of offline RL evaluations for both MDP and POMDP environments, as well as long-horizon planning problems. In all these settings, Decision Stacks matches or significantly outperforms competing approaches while also offering significant flexibility in the choice of generative architectures and training algorithms.

**Limitations and Future Work.** Our experiments are limited to state-based environments and extending Decision Stacks to image-based environments is a promising direction for future work especially in light of the gains we observed for POMDP environments. We are also interested in exploring the benefits of a modular design for pretraining and transfer of modules across similar environments and testing their generalization abilities. Finally, online finetuning of Decision Stacks using techniques similar to Zheng et al. [2022] is also an exciting direction of future work.

# Acknowledgments

This research is supported by a Meta Research Award.

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

# Appendices

# A Model details of Decision Stacks

In Decision Stacks, we employ a modular approach by utilizing separate models for predicting observations, rewards, and actions. These models are trained independently using the technique of teacher forcing [Williams and Zipser, 1989]. This modular design allows us to explore the architectural flexibility of Decision Stacks and incorporate various inductive biases tailored to each modality. Below, we provide a description of the models used for each module in our approach:

## A.1 Observations models

- **Diffusion-based model.** We adopt the diffusion process as a conditional generative model to generate future observations based on current and past observations. Following Decision Diffuser [Ajay et al., 2022], we employ classifier-free guidance [Ho and Salimans, 2022] and low-temperature sampling to generate future observations conditioned on goals. With classifier-free guidance, we first sample $x_K(\tau)$ from a Gaussian noise and refine it to $x_0(\tau)$ with the following procedure in Eq 7 for denoising $x_k(\tau)$ into $x_{k-1}(\tau)$.

$$\hat{\epsilon} := \epsilon_\theta\left(x_k(\tau), \varnothing, k\right) + \omega\left(\epsilon_\theta\left(x_k(\tau), G, k\right) - \epsilon_\theta\left(x_k(\tau), \varnothing, k\right)\right) \qquad (7)$$

where $G$ is the goal information. For the observation model, we diffuse over a sequence of consecutive observations.

$$\tau_{obs} = \begin{bmatrix} o_0 & o_1 & \dots & o_T \end{bmatrix} \qquad (8)$$

We parameterize the diffusion model as U-Net [Ajay et al., 2022, Janner et al., 2022]. To generate future observations, we use an inpainting strategy where we condition the first observation in the diffusion sequence as the current observation $o_0$.

- **Transformer-based model.** The trajectory sequences can also be autoregressively predicted using a transformer architecture as previously demonstrated in Decision Transformer [Chen et al., 2021] and Trajectory Transformer [Janner et al., 2021]. In our approach, we employ a GPT [Radford et al., 2018] model as the underlying transformer architecture. This transformer-based model is trained by feeding it with a sequence of observations, alongside the task goal as the conditioning information. To ensure temporal consistency during both training and testing, we employ causal self-attention masks, enabling the model to predict the next observation solely based on the preceding observations and task information. To effectively represent the observation and task modalities, we treat them as separate entities and embed them into dedicated tokens. These tokens are then input to the transformer, which is further enriched with positional encoding to capture the temporal relationships among the observations. This combination of a GPT model, self-attention masks, and modality-specific token embeddings allows our approach to effectively model the sequential nature of the observations while incorporating task-related information. The transformer can autoregressively generate a sequence of observations as in Eq 8.

## A.2 Reward models

- **Diffusion-based model.** For our reward model, we employ a similar U-Net architecture as the observation diffusion model but diffuse over different sequences. Specifically, we diffuse over the combined sequence of observations and rewards as follows:

$$\tau_{rew} = \begin{bmatrix} o_0 & o_1 & \dots & o_T \\ r_0 & r_1 & \dots & r_T \end{bmatrix} \qquad (9)$$

In this combined sequence, we concatenate the observation sequence $o_0, o_1, ..., o_T$ and the corresponding reward sequence $r_0, r_1, ..., r_T$ at each time step for training. To generate the reward sequence, we utilize an inpainting conditioning strategy, similar to the one used in the observation model. This strategy involves conditioning the diffusion process on the observation sequences $o_0, o_1, ..., o_T$ while generating the rewards. By incorporating this inpainting conditioning, the reward model can effectively utilize the available observation information to generate accurate reward predictions throughout the diffusion process.

- **Transformer-based model.** We employ an Encoder-Decoder transformer architecture for reward sequence generation, given a sequence of observations and the task goal. The encoder module embeds the observations and task goal, incorporating time encoding for capturing temporal dependencies. The encoded inputs are then processed by a transformer layer. The decoder module generates the reward sequence by iteratively predicting the next reward based on the encoded inputs and previously generated rewards. The transformer architecture facilitates capturing long-range dependencies and effectively modeling the dynamics between observations and rewards.
- **MLP reward model.** The multi-layer perceptron (MLP) reward model is a straightforward mapping from the current observation to the corresponding immediate reward. This model does not incorporate context or future synthesized information as it relies on a fixed input and output size. Consequently, the MLP architecture does not consider or incorporate any contextual information during its prediction process.

### A.3 Action models

- **Diffusion-based model.** Similar to the observation and reward diffusion processes, we perform diffusion over the trajectory defined in Equation 10 as well as goal conditioning. By employing diffusion in this manner, we generate the action sequence conditioned on the information contained within the previous observations, rewards, and anticipated future observations and rewards. This diffusion process enables us to effectively capture the dependencies and dynamics among these elements, resulting in the generation of contextually informed action sequences.

$$\tau_{act} = \begin{bmatrix} o_0 & o_1 & \ldots & o_T \\ r_0 & r_1 & \ldots & r_T \\ a_0 & a_1 & \ldots & a_T \end{bmatrix} \tag{10}$$

- **Transformer-based model.** We employ the Encoder-Decoder transformer architecture, similar to that utilized in the reward transformer model, for our action model. The encoder module performs embedding of the observations, rewards, and task goals, incorporating time encodings to capture temporal dependencies. The encoded inputs are subsequently processed by a transformer layer. The decoder module is responsible for generating the action sequence by iteratively predicting the subsequent action based on the encoded inputs and previously generated actions. Consequently, our action model produces coherent and contextually informed action sequences.
- **MLP action model.** The multi-layer perceptron (MLP) action model functions as a direct mapping from the current observation and subsequent observation to the corresponding immediate action, similar to the inverse dynamics model in Decision Diffuser [Ajay et al., 2022]. However, due to its fixed input and output size, this MLP architecture does not incorporate contextual or future synthesized information. Consequently, the model lacks the ability to consider or integrate contextual details. This limitation proves disadvantageous in the context of partially observable Markov decision processes (POMDPs), where the inclusion of contextual information is vital for inferring hidden states and making informed decisions.

## B  Hyperparameters and training details

- The diffusion-based observation model follows the same training hyperparameters as in Decision Diffuser [Ajay et al., 2022], where we set the number of diffusion steps as 200 and the planning horizon as 100.
- For other models, we use a batch size of 32, a learning rate of $3e^{-4}$, and training steps of $2e^6$ with Adam optimizer [Kingma and Ba, 2015].
- The MLP action model and the MLP reward model is a two layered MLP with 512 hidden units and ReLU activations.
- The diffusion models' noise model backbone is a U-Net with six repeated residual blocks. Each block consists of two temporal convolutions, each followed by group norm [Wu and He, 2018], and a final Mish nonlinearity [Misra, 2019].

- For Maze2D experiments, different mazes require different average episode steps to reach to target, we use the planning horizon of 180 for umaze, 256 for medium-maze and 300 for large maze.
- For Maze2D experiments, we use the warm-starting strategy where we perform a reduced number of forward diffusion steps using a previously generated plan as in Diffuser [Janner et al., 2022] to speed up the computation.
- The training of all three models, including the observation, action, and reward models, is conducted using the teacher-forcing technique [Williams and Zipser, 1989].
- Additional hyperparameters can be found in the configuration files within our codebase.
- Upon reproducing the Decision Diffuser (DD) approach [Ajay et al., 2022] using their provided codebase, we observed that the agent performance can be sensitive to the test return of the locomotion tasks, conditional guidance parameter, and sampling noise. For Decision Stacks variants that use the DD observation model, we directly use the models tuned for DD's best performance. In future work, it can be helpful to use conservative regularizers [Nguyen et al., 2022] to further improve both DD and DS performance.
- Each model was trained on a single NVIDIA A5000 GPU.

## C   Sensitivity analysis on the dimension occlusion for POMDPs.

Table 7: **Sensitivity Analysis on Occluded Dimensions for POMDPs**. In the Hopper environment, the full state contains 5 dimensions of position data and 6 dimensions of velocity data of different joints. The table below illustrates experiments with various dimensions and semantics for occlusion. DS consistently exhibits superior or second-best performance in comparison to other baselines on the hopper-medium-expert-v2 dataset. The standard error is reported, and the results are averaged over 15 random seeds.

| Occluded Dimension | BC | DT | TT | DD | Diffuser | DS (ours) |
|---|---|---|---|---|---|---|
| Occlude first 2 dim of velocity | $43.3 \pm 2.2$ | $59.0 \pm 2.3$ | $68.7 \pm 8.3$ | $52.3 \pm 5.5$ | $52.5 \pm 7.4$ | $\mathbf{68.0 \pm 3.4}$ |
| Occlude middle 2 dim of velocity | $34.0 \pm 0.9$ | $73.1 \pm 4.6$ | $76.7 \pm 7.9$ | $31.6 \pm 5.14$ | $24.2 \pm 3.1$ | $\mathbf{81.1 \pm 6.9}$ |
| Occlude last 2 dim of velocity | $51.1 \pm 1.3$ | $105.2 \pm 1.0$ | $61.6 \pm 3.3$ | $32.7 \pm 2.4$ | $70.7 \pm 4.7$ | $\mathbf{110.9 \pm 0.4}$ |
| Occlude first 2 dim of position | $14.6 \pm 2.4$ | $\mathbf{40.4 \pm 3.2}$ | $15.0 \pm 4.1$ | $8.7 \pm 0.2$ | $29.5 \pm 1.7$ | $38.9 \pm 6.2$ |
| **Average** | $35.7 \pm 1.7$ | $69.4 \pm 2.8$ | $55.5 \pm 5.9$ | $31.3 \pm 3.3$ | $44.2 \pm 4.2$ | $\mathbf{74.7 \pm 4.2}$ |

## D   Ablation of reward modeling across MDP and POMDP.

We performed an ablation study on reward modeling, focusing on six locomotion environments. We trained two sets of transformer-based action models: one set modeling the sequence of observations and actions, and the other set incorporating the sequence of observations, rewards, and actions. The evaluation of these models was conducted on both Markov Decision Process (MDP) and Partially Observable Markov Decision Process (POMDP) environments, specifically three locomotion tasks. The results obtained from this study highlight the evident advantages of reward modeling, particularly in dense-reward locomotion tasks. The details and findings are presented in the Table 8, emphasizing the benefits gained from incorporating reward modeling.

Table 8: Ablation on reward modeling.

| Environments | | Action models | |
| | | Transformer **with** reward modelling | Transformer **without** reward modelling |
|---|---|---|---|
| MDP | Hopper-medium-v2 | 76.6 ±4.2 | 69.7 ±3.4 |
| | Walker2d-medium-v2 | 83.6 ±0.3 | 82.7 ±1.2 |
| | Halfcheetah-medium-v2 | 47.8 ±0.4 | 43.0 ±2.8 |
| | Average (MDP envs) | **69.3** | 65.1 |
| POMDP | Hopper-medium-v2 | 56.0 ±3.5 | 43.6 ±1.3 |
| | Walker2d-medium-v2 | 74.3 ±4.2 | 54.2 ±7.3 |
| | Halfcheetah-medium-v2 | 47.1 ±0.3 | 46.5 ±0.7 |
| | Average (POMDP envs) | **59.1** | 48.1 |
| | Average (All envs) | **64.2** | 56.6 |

# E  Modeling ordering ablation study

Table 9: **Performance Comparison on Modeling Ordering.** From the chain rule of probability, any autoregressive factorization can model the data distribution under idealized conditions. In practice, we choose the ordering of observations, rewards, and actions. Specifically, we ordered observations prior to rewards to be consistent with the functional definitions in MDPs, where the reward is typically a function of observations (and potentially other variables), but not vice versa. We tested this choice empirically as well in early experiments, and found our choice to significantly outperform the counterpart. The table provides a comparison of two different orderings: Reward-State-Action (R, S, A) and State-Reward-Action (S, R, A). The results indicate that the S, R, A ordering outperforms the R, S, A ordering in the halfcheetah-medium-replay-v2 environment. DS models observations prior to rewards to be consistent with the functional definitions in MDPs, where the reward is typically a function of observations (and potentially other variables), but not vice versa.

| Ordering | halfcheetah-medium-replay-v2 Performance |
|---|---|
| R, S, A | $32.1 \pm 0.1$ |
| S, R, A | **$41.1 \pm 0.1$** |

# F  Maze2D additional analysis

We investigate trajectory sampling-based methods, namely DD, Diffuser, and Decision Stacks, within this specific environment. All three models utilize a diffusion-based approach to generate future plans. In this task with goal-conditioning, the position of the maze's goal serves as the condition.

Regarding Diffuser, we strictly adhere to their codebase and observed that Diffuser impaints the last goal position into the diffusion sequence. Similarly, for DD, we follow their codebase and incorporate the goal as conditioning, embedding it and transmitting it to the diffusion backbone, which is an Unet. In the case of DD, we also apply inpainting conditioning by conditioning the last position in the diffusion sequence to serve as the goal position.

## F.1  Detour problem of inpainting conditioning

However, we encountered an issue with this inpainting goal conditioning when performing open-loop generation. The problem arises because, during each replanning iteration of the diffusion model, it only conditions on the last time step to be the goal state. Consequently, when the agent is in close proximity to the goal point, the diffusion model plans a detour that initially takes the agent away from the goal and then redirects it back towards the goal. Since the environment's reward function is designed in such a way that the agent receives a reward when it is near the goal, and the episode does

not terminate upon reaching the goal point but rather when it reaches the maximum episode length, it is actually more advantageous for the agent to remain at the goal point.

After recognizing this issue, we devised a "progressive conditioning" (PC) strategy to address it. This approach involves gradually increasing the number of timesteps in the diffusion sequence that are conditioned to be the goal position as time progresses. By implementing this progressive conditioning method, we successfully resolve the detour problem and provide additional incentives for the agents to remain at the goal position. Without PC, the agent's behavior during open-loop evaluation exhibits a recurring pattern of moving toward the goal and deviating from it. This results in unfavorable trajectories where the agent fails to stay at the goal position.

The integration of progressive conditioning led to performance enhancement in both Diffuser and DD, although the overall results still indicate that Decision Stacks outperforms DD in both single goal (average performance: 109.8 vs. 131.5) and multi goal environments (average performance: 111.6 vs. 123.4). We compare the performance of Diffuser, DD, DS with and without progressive conditioning in the bar charts below in Figure 4 and Figure 5. The results indicate that DS with PC outperforms other baselines on most of the environment settings across single-goal and multi-goal settings.

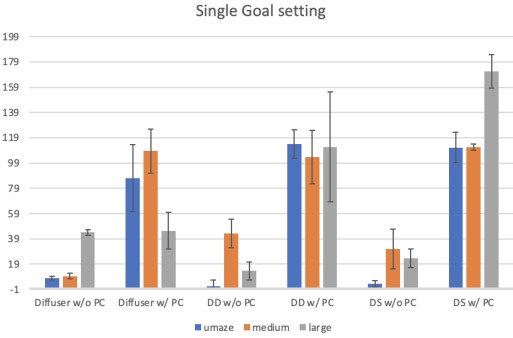

Figure 4: Bar chart comparison of Diffuser, DD and DS with and without progressive conditioning in the single goal setting of Maze2D.

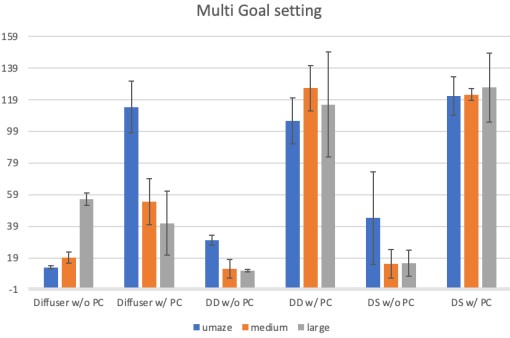

Figure 5: Bar chart comparison of Diffuser, DD and DS with and without progressive conditioning in the multi goal setting of Maze2D.

