# OpenReview forum: "Decision Stacks: Flexible Reinforcement Learning via Modular Generative Models"
_NeurIPS.cc/2023/Conference — NeurIPS 2023 poster_

### Official Review · Reviewer_ywNv · 2023-06-28

**Soundness:** 2 fair
**Presentation:** 3 good
**Contribution:** 2 fair
**Rating:** 3
**Confidence:** 4

**Summary:**

This paper focuses on solving offline RL with generative models. Concretely, it proposes a method to modularize the joint distribution of time-induced trajectories and use separate generative models to represent observation module, reward module, and action module. Evaluations are conducted on D4RL benchmark with MDP and POMDP environments. Extensive comparisons against prior works are included.

**Strengths:**

- The paper is fairly well presented and easy to follow.

- The proposed method is extensively evaluated against multiple related approaches.

**Weaknesses:**

- Novelty of the proposed method is limited. The improved performance can also be attributed to other confounding factors, such as larger models due to reward and action modules being factored out.

- It is counter-intuitive to ignore the canonical time-induced casual ordering in favor of different token types. More intuitions and theoretical analysis (if applicable) are encouraged to provide.

- Lack of evidences to support the claims on modular expressivity (L145). Are there any experiments showing it can transfer to new environments?

- The empirical results, especially those on D4RL locomotion tasks, are not significant enough to justify the extra compute introduced.

- Just swapping each module with different model architectures/modeling strategies is not enough to gain insights of the proposed method. More ablations and analysis are necessary.

**Questions:**

- Why DS's performance on POMDP Hopper task with medium-expert data (Table 3) is even better than that on fully observable setting (Table 2)?

- How does DS compare to more straightforward baseline, such as the one with separate heads for three modules but shares the same backbone?

**Limitations:**

- The design choice of ignoring temporal causality is not well motivated. Case studies or even proof-of-concept experiments would be helpful to justify it.

- The performance improvement could be attributed to confounding factors such as larger model sizes. More experiments with controlled variables (e.g., same amount of model parameters) would help to support claims made in the paper.

---

> ### Author Rebuttal · Authors · 2023-08-10
>
> Thanks for the valuable comments and very detailed insights! We address your questions below.
>
> > Q1: “Novelty of the proposed method is limited. The improved performance can also be attributed to other confounding factors, such as larger models due to reward and action modules being factored out. “
>
> We hope the following clarify DS's novel modular and expressive design:
>
> 1. The performance on D4RL Mujoco MDPs are saturated with a number of recent methods performing quite close. While our gains are close, the fact that DS is at the top serves as a promising sanity check and indication of its strong performance.
>
> 2. We developed more challenging POMDP environment by excluding two velocity dimensions. During this rebuttal period, new experiments were conducted for dimension occlusion sensitivity analysis. The results, shown in Table 1 in the rebuttal PDF, strengthens DS's superior performance in POMDP cases.
>
> 3. On the Maze2D environment, we showcase the planning ability of DS. This is the key environment used for evaluating planning capabilities in our most closely related works, DD and Diffuser. The superior performance in this environment relative to baselines illustrates the strengths of our method in complex single and multi-goal planning tasks.
>
> 4. New experiments were conducted to showcase the compositional generalization of DS brought by the decomposed modules, as in Table 2 in the rebuttal PDF, where the same observation and/or reward models can be reused for different environments in Maze2D.
>
> 5. DS's modular design is less complex as it explicitly respects the different modalities for states, actions, rewards. For example, actions can be less continuous than the states. Using a single diffusion or transformer model to model them together requires adjusting for modality-specific encoders and decoders, as well as close monitoring of training dynamics.
>
>
> > Q2: “counter-intuitive to ignore the canonical time-induced casual ordering in favor of different token types. More intuitions and theoretical analysis (if applicable) are encouraged to provide.” “The design choice of ignoring temporal causality is not well motivated. Case studies or even proof-of-concept experiments would be helpful to justify it.”
>
> DS introduces a novel token-first ordering of the trajectory components instead of the timestep-first arrangements. This design, while appearing counter-intuitive, serves 2 important functions:
>
> (1)  A modality-focused approach explicitly emphasizes the role of planning. Temporal ordered models such as DT do not permit planning.
>
> (2) It caters to the distinct semantic attributes of observations, actions, and rewards. It recognizes the unique characteristics such as dimensionality and domain type (discrete or continuous). By allowing for separate modeling, DS provides flexibility in architectural choices to cater to these differences, unlike other methods such as DT, TT and Diffuser that chain states, actions, (and rewards) together. The promising results in our experiments, especially on the POMDPs and Maze2D have validated DS with superior expressivity and planning ability. This modular design also brings novel compositional generalizations.
>
> >Q3: “Lack of evidences to support the claims on modular expressivity (L145)...any experiments showing it can transfer to new environments?”  “Just swapping each module with different model architectures/modeling strategies is not enough to gain insights of the proposed method. More ablations and analysis are necessary.”
>
> In line with your suggestion, we conducted new experiments to explore the potential of our decomposed submodules for compositional generalization and efficient multitask learning. Specifically, we focus on the ability to transfer to other environments using shared observation models. We created modified versions of the Maze2D environment with varying action and reward spaces (details in Table 2 caption), maintaining the same observation model. Our results demonstrate that DS facilitates an efficient compositional generalization in scenarios where reusable components exist across different environments as detailed in Table 2 in the rebuttal PDF.
>
> >Q4: “The empirical results, especially those on D4RL locomotion tasks, are not significant enough to justify the extra compute introduced.”
>
> 1. We justified the significance of our results in Q1.
>
> 2. No significant extra compute: Each module is parameterized using the same kind of architecture (transformers or diffusion) as the other baselines. The training time for DS does not increase as compared to the closest baseline DD despite having three generative models, due to DS's parallel module training. The training time is determined by the state model of DS due to high dimensionality. A training cost comparison in Table 4 of the rebuttal PDF confirms that DS's training cost aligns with other baselines.
>
> >Q5: Why DS's performance on POMDP Hopper task with medium-expert data is even better than that on fully observable setting?
>
> The performances in POMDP and MDP settings are similar within the error bar range. The higher performance in the POMDP might stem from the limited 15 rollouts of a random seed. This suggests that DS is capable of extracting enough useful information to infer hidden states. This trend is not unique to DS, e.g., DT's performance on the Hopper-Medium-Expert task is close in both the POMDP and MDP cases.
>
> >Q6: How does DS compare to more straightforward baseline, such as the one with separate heads for three modules but shares the same backbone?
>
> This baseline resembles the Trajectory Transformer, which predicts the states, actions, and rewards with a common backbone architecture. In comparison, DS attains performance gains of 2.7% on MDPs and 40.2% on POMDPs over TT. Further, compared to such baselines, DS's decomposed modules bring more architectural flexibility (Table 4) and transferrability across environments (Table 2, rebuttal pdf).

---

> > ### Author Response · Authors · 2023-08-14
> > **Reminder**
> >
> > Thank you again for your valuable feedback on our paper. We've carefully considered your comments and conducted new experiments to address the weaknesses and questions you highlighted. We'd like to note that the discussion phase is drawing to a close, and we hope you've had a chance to review our response. Given the limited time, are there any other questions or concerns you'd like us to address? We would greatly appreciate your support in this regard.

---

> > > ### Comment · Reviewer_ywNv · 2023-08-15
> > >
> > > Thanks authors for attempting to address my questions. However, I'm still concerned about the following aspects.
> > >
> > > - Uniqueness of the proposed method (also mentioned by reviewer Xfrm);
> > > - Lack of rational justification of design choices, including increased model complexity (also mentioned by reviewers Xfrm Q4 and xgrf) and the neglection of temporal dependency and casual dependency between three random variables (also mentioned by reviewer xgrf);
> > > - The relatively simple and toyish evaluation tasks and superficial construction of the POMDP setting (also mentioned by reviewer E8pR);
> > > - Absence of arguments or proofs that support claims made in this paper (also mentioned by reviewer vCY4). For example:
> > > 1) L5 and L148 "maximal expressivity". It's possible that there is other modeling strategy that can be more expressive. Authors neither theoretically justify this, nor empirically demonstrate this.
> > > 2) L10 "our framework guarantees both expressivity and flexibility". The word choice of "guarantee" is inappropriate due to the lack of rigorous proof.
> > > - Lack of in-depth analysis into the flexibility provided by this model. Conducting experiments on all the Cartesian product {Observation Models} $\times$ {Action Models} $\times$ {Reward Models} is respectful, insightful conclusions/takeaways are missed though.
> > >
> > > Furthermore, I have the following questions regarding authors' reply:
> > > - Regarding table 2, which one from the three maze environments is modified? Are the transferred reward model and observation model kept frozen during training (otherwise the seemingly good transfer results could be attributed to the joint training in new tasks)? With the transferred reward model and observation model, does the experiment achieve better sample efficiency? Modifications on the action space are far-fetched to be called new tasks.
> > > - If the better performance on Hopper POMDP may be explained by the limited number of rollouts, wouldn't this suggest that experiment results reported in this work may not be statistically significant, also due to the limited evaluation rollouts?
> > > - FLOPS is usually used to measure computation. Regarding table 4, what are FLOPS values for DS and baselines?

---

> > > > ### Author Response · Authors · 2023-08-17
> > > > **response (1/2)**
> > > >
> > > > We thank the reviewer for the response.
> > > > >Uniqueness of the proposed method (also mentioned by reviewer Xfrm);
> > > >
> > > > We have addressed this aspect in our original rebuttal. Given the subjective nature of this follow-up comment, can you elaborate what is the remaining issue? Also, note the comment by reviewer Xfrm is in a different context in comparison with a specific form for trajectory transformer, which we also addressed in our rebuttal response for the reviewer.
> > > >
> > > > >Lack of rational justification of design choices, including increased model complexity (also mentioned by reviewers Xfrm Q4 and xgrf) and the neglection of temporal dependency and casual dependency between three random variables (also mentioned by reviewer xgrf);
> > > >
> > > > We have addressed these questions in our rebuttal. To reiterate, we respectfully refute the comment that there is an increase in training time. We also elaborated on why temporal dependency is not necessary due to the chain rule and also ignores the distinct modality attributes. For both, we also included empirical evidence validating our design choices. We would be grateful if you can specify why our response in the original rebuttal does not address your original question.
> > > >
> > > > >The relatively simple and toyish evaluation tasks and superficial construction of the POMDP setting (also mentioned by reviewer E8pR);
> > > >
> > > > We respectfully disagree that our evaluation tasks are “toyish”. D4RL is the standard benchmark used in all major offline RL papers in the last few years [CQL, DT, IQL, Diffuser, DD] . Maze2D task was used for benchmarking planning by Diffuser (published at NeurIPS 2021) and DD (published at ICLR 2023, Oral), which are two of the closest baselines to our method. We've undertaken a comprehensive sensitivity analysis that varies the occluded dimensions for the POMDPs scenario as in Table 1 in the rebuttal PDF. Additionally, it's worth noting that our POMDPs construction on Mujuco tasks, specifically the case of exclusion of all velocity dimensions, was used as the main experimental setting in https://arxiv.org/pdf/1906.09510.pdf (UAI 2019) and also used in https://arxiv.org/pdf/1604.06778.pdf (ICML 2016) sec 3.3.
> > > >
> > > >
> > > > >Absence of arguments or proofs that support claims made in this paper (also mentioned by reviewer vCY4). For example:
> > > > L5 and L148 "maximal expressivity". It's possible that there is other modeling strategy that can be more expressive. Authors neither theoretically justify this, nor empirically demonstrate this.
> > > > L10 "our framework guarantees both expressivity and flexibility". The word choice of "guarantee" is inappropriate due to the lack of rigorous proof.
> > > >
> > > > To reiterate our response to reviewer vCY4, maximal expressivity is guaranteed from our autoregressive factorization in Eq. 3. That is, since autoregressive models are derived from chain rule of probability, under idealized assumptions on model expressivity and dataset size, they can learn any data distribution. For flexibility, we refer to the property that the modular design allows us to use any architecture and training objective for the observation, reward, and action modules. As we mentioned to reviewer vCY4, we will edit and clarify the language in the final version.
> > > >
> > > > >Lack of in-depth analysis into the flexibility provided by this model. Conducting experiments on all the Cartesian product … is respectful, insightful conclusions/takeaways are missed though.
> > > >
> > > >
> > > > Below we list the key insights from DS’s flexibility:
> > > >
> > > > Flexibility in architecture parameterization: Through our architectural flexibility experiments, we've showcased that each of the three modules can be parameterized using any appropriate architecture.
> > > >
> > > > Compositional transfer: To further illustrate the versatility of DS, we've conducted compositional generalization experiments. These demonstrate that observation and reward models can be effectively reused (while remaining frozen) for environments with varying action spaces.
> > > >
> > > > If you are looking for a specific kind of insight, please let us know and we’d be happy to discuss it.

---

> > > > > ### Author Response · Authors · 2023-08-17
> > > > > **response (2/2)**
> > > > >
> > > > > >Regarding table 2, which one from the three maze environments is modified? Are the transferred reward model and observation model kept frozen during training (otherwise the seemingly good transfer results could be attributed to the joint training in new tasks)? With the transferred reward model and observation model, does the experiment achieve better sample efficiency? Modifications on the action space are far-fetched to be called new tasks.
> > > > >
> > > > > Regarding Table 2, the experiments were carried out in the Maze2d-Umaze-v2 environment.
> > > > > Yes, both the reward and observation models remain frozen during transfer to different action spaces. The one observation model is shared across 6 environments; each of the two reward models is reused across 3 action spaces. By reusing these models across varying reward and action spaces, we can sidestep the need for their retraining, leading to more modular training compared to finetuning the whole model and better performance.
> > > > >
> > > > > >If the better performance on Hopper POMDP may be explained by the limited number of rollouts, wouldn't this suggest that experiment results reported in this work may not be statistically significant, also due to the limited evaluation rollouts?
> > > > >
> > > > > We used a similar or more number of rollouts than prior studies and this should not be diminishing the statistical significance of our method. Additionally, our broader sensitivity analysis consistently demonstrated the better performance of DS across multiple POMDP environments, showcasing its robust performance.
> > > > >
> > > > > >FLOPS is usually used to measure computation. Regarding table 4, what are FLOPS values for DS and baselines?
> > > > >
> > > > > Thank you for the suggestion. Our FLOPs are comparable to those of the closest baseline DD (3618 vs. 3287). However, we also note that FLOPs primarily account for computational operations and tend to overlook other integral aspects of real-world processing speed such as data transfer, I/O operations, and varied latencies [1]. As a result, even algorithms with matching FLOPs can deliver different performances across diverse hardware architectures, especially when certain hardware elements are optimized for specific operations. That is why we reported wall-clock time in our original experiments. This metric offers a comprehensive view of real-world performance, capturing both computational and other related delays, thus providing a direct insight into algorithmic efficiency.
> > > > >
> > > > > Also we emphasize that in our comparisons, we did not intentionally select any baseline to appear smaller, either in size or FLOPs. Every method is tailored to harness the maximum potential of the architecture it operates on and the architectures have been tuned extensively by the original authors of each baseline method. In fact, a method's inability to deliver optimal performance on larger architectures often signals limitations in scalability.
> > > > >
> > > > > [1] Bartoldson et al. Compute-Efficient Deep Learning: Algorithmic Trends and Opportunities. JMLR. https://arxiv.org/abs/2210.06640

---

> > > > > > ### Author Response · Authors · 2023-08-18
> > > > > > **A broader sensitivity analysis on the construction of the POMDP setting**
> > > > > >
> > > > > > Dear reviewer ywNv,
> > > > > >
> > > > > > Thank you for engaging with us during the discussion phase! In our discussions with reviewer Xfrm, we conducted a broader sensitivity analysis of the excluded dimension within the POMDP experiments, as detailed in the table below. We believe this result helps address the question you raised below:
> > > > > >
> > > > > > > superficial construction of the POMDP setting
> > > > > >
> > > > > > By testing varying numbers of occluded dimensions and different semantics, such as velocity and position, we found that DS consistently outperforms other baselines in overall performance.
> > > > > >
> > > > > > | Occluded dimension                     | BC       | DT        | TT        | DD       | Diffuser | DS(ours)  |
> > > > > > |---------------------------------------|----------|---------- |---------- |----------|----------|---------- |
> > > > > > | Occlude first 2 dim of velocity       | 43.3±2.2 | 59.0±2.3  | **68.7±8.3**  | 52.3±5.5 | 52.5±7.4 | 68.0±3.4 |
> > > > > > | Occlude middle 2 dim of velocity      | 34.0±0.9 | 73.1±4.6  | 76.7±7.9  | 31.6±5.1 | 24.2±3.1 | **81.1±6.9** |
> > > > > > | Occlude last 2 dim of velocity        | 51.1±1.3 | 105.2±1.0| 61.6±3.3 | 32.7±2.4 | 70.7±4.7 | **110.9±0.4**|
> > > > > > | Occlude last 4 dim of velocity        | 2.5±0.1  | 53.6±4.8  | 59.3±6.8  | 9.0±2.6  | 5.8±1.7  | **69.0±9.9** |
> > > > > > | Occlude last 6 dim of velocity        | 3.4±0.1  | 30.4±4.9  | **57.9±5.2**  | 3.8±8.1  | 2.9±0.2  | 33.0±4.7|
> > > > > > | Occlude first 2 dim of position       | 14.6±2.4 | **40.4±3.2**  | 15.0±4.1  | 8.7±9.3  | 29.5±1.7 | 38.9±6.2 |
> > > > > > | Occlude first 5 all dim of position   | 10.4±1.0 | 27.6±5.3  | 38.9±4.1  | 10.7±0.8 | 12.4±3.7 | **39.0±2.5** |
> > > > > > | Average                               | 22.7±1.1 | 55.6±3.7  | 54.0±5.7  | 21.3±4.8 | 28.3±3.2 | **62.8±4.9** |
> > > > > >
> > > > > > As it approaches the discussion deadline, we sincerely hope you've had an opportunity to review our response. May we kindly inquire if there are any additional questions or concerns you'd like us to address? We truly value your feedback and insights. Thank you!

---

### Official Review · Reviewer_xgrf · 2023-07-03

**Soundness:** 3 good
**Presentation:** 3 good
**Contribution:** 2 fair
**Rating:** 4
**Confidence:** 4

**Summary:**

This paper highlights a drawback in prior frameworks, such as Decision Transformer and Diffuser, where the absence of modular hierarchies among different tokens results in limited expressivity and flexibility. To overcome this issue, the paper introduces Decision Stacks (DS), a modular algorithm designed for learning goal-conditioned policies using offline datasets. The proposed method parameterizes three generative model-based modules for future observation prediction, reward estimation, and action generation, respectively. Through various offline evaluations in both MDP and POMDP environments, this paper demonstrates that DS outperforms previous approaches by generating superior plans.

**Strengths:**

This paper presents a modular probabilistic framework, utilizing deep generative models to establish token-level hierarchies in trajectory generation. The proposed algorithm, Decision Stacks (DS), incorporates independent generative models for simulating the temporal evolution of observations, rewards, and actions, allowing for parallel learning and enabling flexible generative decision making. The algorithm is both straightforward and effective, particularly in the POMDP setting, where DS surpasses other baseline methods by a significant margin. Additionally, the experiments are conducted meticulously, and the visualizations of example rollouts in the Maze2D-medium-v1 environment are clear.

**Weaknesses:**

One major concern regarding the proposed algorithm DS is the high complexity of the model, as it requires training three generative models. However, in most experiments such as offline RL with an MDP setting, the performance improvement compared to other baselines that use only one generative model is not significant. Furthermore, the paper lacks an explanation of the specific dimensions that were excluded to construct the POMDP setting. Additionally, as depicted in Figure 1, the generation of observation, reward, and action sequences follows a sequential order, and there exist dependencies among the three generative models. However, the paper does not provide clear explanations on how training can be parallelized and how the choice of generative models affects performance.


**Questions:**

(1)How is the handcoded controller implemented in 4.1 and 4.2, respectively? What role does it play? In Table 1 and Figure 1, why do Decision Diffuser (DD) and DS exhibit similar performance when this controller is added? DD does not adopt a modular structure but utilizes the same diffusion-based observation model as DS. Does this indirectly suggest that the modular structure has minimal impact as long as a good generative model is chosen?
(2)During the experiment, why was a diffusion model used for the observation model while transformer models were used for the action and reward models? Why do different choices of generative models lead to the results shown in Table 4? It is hoped that the author can provide corresponding reasonable explanations instead of simple combination attempts.

typoes, e.g., "seggregate" --> "segregate".

**Limitations:**

Methods such as Decision Transformer [1], which reduce reinforcement learning to a prediction task, have gained popularity partly due to their simplicity. In contrast, DS trades a high model complexity for relatively modest expressivity. It would be more convincing in terms of expressivity if experiments were conducted in stochastic environments[2] that yield better results. Furthermore, the paper lacks rational analysis in certain aspects, such as the selection of generative models to generate higher quality trajectories.
[1] Chen L, Lu K, Rajeswaran A, et al. Decision transformer: Reinforcement learning via sequence modeling[J]. Advances in neural information processing systems, 2021, 34: 15084-15097.
[2] Paster K, McIlraith S, Ba J. You Can't Count on Luck: Why Decision Transformers Fail in Stochastic Environments[J]. arXiv preprint arXiv:2205.15967, 2022.

---

> ### Author Rebuttal · Authors · 2023-08-10
>
> Thanks for the valuable comments and very detailed insights! We address your questions below.
> >Q1: “One major concern regarding DS is the high complexity, as it requires training three generative models...the performance improvement compared to other baselines that use only one generative model is not significant.”
>
> We respectfully disagree with the concerns about the complexity and performance of DS. On the contrary, we argue DS is simpler and more effective.
>
> Complexity:
>
> 1. Each module is parameterized using the same kind of architecture (transformers or diffusion model) as the other baselines. Training time for DS doesn't increase compared to the close baseline DD, despite using three generative models. This is because DS's modularity allows for the parallel training of the three modules. The training time is mainly influenced by the state model of DS due to its high dimensionality. A training cost comparison in Table 4 of the rebuttal PDF confirms that DS's training cost aligns with those of DT, TT, DD, and Diffuser.
>
> 2. Enhanced modality-specific flexibility: Although baselines like DT and TT which use a single generative model, bring surface-level simplicity, we believe DS's modular design is less complex as it explicitly respects the different modalities for states, actions, rewards. For example, actions can be less continuous than the states. Using a single diffusion or transformer model to model them together requires adjusting for modality-specific encoders and decoders, as well as close monitoring of training dynamics.
>
>
> Performance Gain:
>
> 1. In the MDP setting where benchmarks are nearly saturated as recent methods such as DD and TT achieve similar results, DS still performs better, which serves as an essential sanity check and indicates its strong empirical performance, which we validated with our subsequent experiments.
>
> 2. DS demonstrates robust superiority in more complex environments, such as the POMDP setting and the Maze2D environment. The improvements over other methods in these settings are more substantial, indicating DS’s expressive modeling derived from chain rules and D-separation rule which can be applied to the graphical model in Fig 2 translate into tangible performance gains in more complex tasks.
>
> 3. DS modular design allows efficient compositional generalization. We demonstrate this on maze2D where modules can be reusable across envs, as shown in Table 2 of the rebuttal PDF.
>
>
> >Q2: lacks an explanation of the specific dimensions that were excluded in the POMDP setting.
>
> In Section 4.3, we excluded two specific velocity dimensions of the joints from the full state representation for all locomotion environments, simulating lacking relevant sensors. We appreciate your inquiry into this design choice and conducted a sensitivity analysis on dimension occlusions in Table 1 of the rebuttal PDF. The new analysis considered various exclusions and the results affirm that DS continues to perform robustly, outperforming other baselines.
>
> > Q3: “... how training can be parallelized and how the choice of generative models affects performance.”
>
> DS modules’ parallel training is achieved by a technique known as teacher forcing, where during training, the true previous output is fed as input to the next time step. We leverage the computational efficiency of parallel processing without losing the integrity of the dependencies among the models.
>
> >Q4: “ why was a diffusion model used for the observation model while transformer models were used for the action and reward models? Why do different choices of generative models lead to the results in Table 4? … provide corresponding reasonable explanations instead of simple combination attempts.” “... lacks rational analysis in the selection of generative models to generate higher quality trajectories. “
>
> In our experiment, we chose models aligned with the properties of the data modality: diffusion models for the continuous state sequences, and transformers for the more high-frequency action and reward sequences. This wasn't a combination attempt. For action sequences, which often appear as joint torques and tend to be more high-frequency and less smooth, we selected the transformer model. The same applies to our choice of the reward model. This decision was based on the transformer's suitability for handling such high-frequency and varied data structures, as evidenced by its superior performance in Table 4 (55.2 vs. 53.5 for the diffusion action model). By aligning the models with the nature of the data, we optimized our architecture to reflect the unique characteristics of the modalities in RL environments.
>
> >Q5: "...handcoded controller implemented in 4.1 and 4.2? ... role? In Table 1 and Figure 1, why do Decision Diffuser (DD) and DS exhibit similar performance when this controller is added? DD does not adopt a modular structure but utilizes the same diffusion-based observation model as DS. Does this indirectly suggest that the modular structure has minimal impact as long as a good generative model is chosen?"
>
> The handcoded controller is an existing technique used in Diffuser, which calculates action based on planned waypoints with the physics dynamics. It computes the target distance and optimizes actions. Though DD and DS perform similarly with this controller, the key distinction lies in executing plans with action models. Diffuser overlooks the varied data nature between states and actions by diffusing them together, while DD uses an MLP dynamics model that does not consider sufficient contexts. DS employs a more expressive approach with more contextual information. While planning in DD and DS may appear similar, the execution varies, and DS shows superior performance due to its modular structure, especially in the nuanced execution phase. The modular structure of DS does indeed have a significant impact, specifically in the more nuanced execution phase, as opposed to merely selecting a good generative model.

---

> > ### Author Response · Authors · 2023-08-18
> > **Reminder 2 + additional experiments on POMDP settings**
> >
> > Thank you again for your valuable feedback on our paper. We've carefully considered your comments and conducted new experiments to address the weaknesses and questions you highlighted.
> >
> > In our discussions with reviewer Xfrm, we conducted a broader sensitivity analysis of the excluded dimension within the POMDP experiments beyond our original rebuttal as well, as detailed in the table below. We believe these findings also help address the question you raised.
> >
> > > Q2: the paper lacks explanation regarding specific dimensions excluded in the POMDP setting is missing.
> >
> > By testing varying numbers of occluded dimensions and different semantics, such as velocity and position, we found that DS consistently outperforms in overall performance.
> >
> > | Occluded dimension                     | BC       | DT        | TT        | DD       | Diffuser | DS(ours)  |
> > |---------------------------------------|----------|---------- |---------- |----------|----------|---------- |
> > | Occlude first 2 dim of velocity       | 43.3±2.2 | 59.0±2.3  | **68.7±8.3**  | 52.3±5.5 | 52.5±7.4 | 68.0±3.4 |
> > | Occlude middle 2 dim of velocity      | 34.0±0.9 | 73.1±4.6  | 76.7±7.9  | 31.6±5.1 | 24.2±3.1 | **81.1±6.9** |
> > | Occlude last 2 dim of velocity        | 51.1±1.3 | 105.2±1.0| 61.6±3.3 | 32.7±2.4 | 70.7±4.7 | **110.9±0.4**|
> > | Occlude last 4 dim of velocity        | 2.5±0.1  | 53.6±4.8  | 59.3±6.8  | 9.0±2.6  | 5.8±1.7  | **69.0±9.9** |
> > | Occlude last 6 dim of velocity        | 3.4±0.1  | 30.4±4.9  | **57.9±5.2**  | 3.8±8.1  | 2.9±0.2  | 33.0±4.7|
> > | Occlude first 2 dim of position       | 14.6±2.4 | **40.4±3.2**  | 15.0±4.1  | 8.7±9.3  | 29.5±1.7 | 38.9±6.2 |
> > | Occlude first 5 all dim of position   | 10.4±1.0 | 27.6±5.3  | 38.9±4.1  | 10.7±0.8 | 12.4±3.7 | **39.0±2.5** |
> > | Average                               | 22.7±1.1 | 55.6±3.7  | 54.0±5.7  | 21.3±4.8 | 28.3±3.2 | **62.8±4.9** |
> >
> > We'd like to note that the discussion phase is drawing to a close, and we hope you've had a chance to review our response. Given the limited time, please let us know if there are there any other questions or concerns you'd like us to address.

---

> > ### Comment · Reviewer_xgrf · 2023-08-19
> >
> > Thank you for your response and new experimental results.
> > （1）Regarding question 4, the author has provided a relatively brief analysis of data modalities. However, there seems to be a lack of in-depth analysis and discussion about the specific structural advantages of the diffusion model and the transformer in handling these data modalities. Furthermore, there appears to be no in-depth discussion regarding the results generated by different model combinations in Table 4.
> > （2）Tackling serialized trajectory information aids in inferring missing state information, which is a common practice in POMDPs. As mentioned in the Limitations section, the expressiveness in stochastic environments may be more challenging and persuasive compared to POMDP settings.

---

> ### Author Response · Authors · 2023-08-14
> **Reminder**
>
> Thank you again for your valuable feedback on our paper. We've carefully considered your comments and conducted new experiments to address the weaknesses and questions you highlighted. We'd like to note that the discussion phase is drawing to a close, and we hope you've had a chance to review our response. Given the limited time, are there any other questions or concerns you'd like us to address? We would greatly appreciate your support in this regard.

---

### Official Review · Reviewer_E8pR · 2023-07-05

**Soundness:** 3 good
**Presentation:** 3 good
**Contribution:** 3 good
**Rating:** 7
**Confidence:** 4

**Summary:**

This paper proposes to tackle the problem decision making using a stack of different generative models. The first generative model constructs a conditional distribution over observations. A subsequent generative model constructs a conditional distribution over rewards, with a final generative model constructs a distribution over actions. The authors illustrate the efficacy of this decomposed approach across a suite of different tasks.

**Strengths:**

- I enjoyed reading the paper -- it was quite clear and easy to read. The motivation of the paper to decompose the generative modeling objective into a set of component modules is sound.
- The paper follows a formulation of offline reinforcement learning as probabilistic inference that is likely to relevant to a large audience at NeurIPS with the increasing popularity of generative models
- The approach performs well across a set of different environments, outperforming existing baselines across 3 separate tasks.

**Weaknesses:**

- The results illustrated in the approach are a bit toy -- with the largest gains in the Maze2D environment. The Mujoco control environment has somewhat limited gains and the constructed POMDP environment seems a bit artificial. It may be that the Mujoco control environments have already saturated in performance and it may be interesting to try more complex planning tasks such as RLBench or other robotic manipulation environments.
- While I understand the motivation to decompose trajectory synthesis into a set of modular components, I'm not sure I completely understand the particular decomposition of first observations, then rewards, and then actions. It seems like decomposing rewards first may be more natural then observations.
- It might also be interesting to explore the extent to which decomposed submodules can enable compositional generalization or more efficient multitask learning by encoding structure in the greneration procedure. For instance -- perhaps the observation model can be used across a set of different tasks.


There are a couple of typos in the paper listed below:

- L133 typo auotregressive -> autoregressive
- In L303  Dai et al should be Du et al. The method does not use a state model as a pretrained text2image model but rather a learned text2video model.
- The table before section 4.2 is misformatted and extends to the margin of the paper

**Questions:**

I had a couple questions about the underlying approach that might be interesting to discuss:

- In equation 6, we sequentially sample from each factored distribution to sample from equation 3. In practice, would we instead want to sample marginalized distribution over actions across all possible observations and rewards given the goal?
- It might be interesting to think a bit about how the individual decomposed modules can also provide feedback with respect to each other. For instance a planned set of states may induce a poor distribution over rewards. Having the conditional reward model then provide some feedback to regenerate the planned set of states may then be interesting. [1] may be an interesting read

[1] Composing Ensembles of Pretrained Networks via Iterative Consensus

**Limitations:**

Yes

---

> ### Author Rebuttal · Authors · 2023-08-10
>
> Thank you for your valuable comments and very detailed insights! We address your questions below.
>
> >Q1: The results illustrated in the approach are a bit toy -- with the largest gains in the Maze2D environment. The Mujoco control environment has somewhat limited gains and the constructed POMDP environment seems a bit artificial. It may be that the Mujoco control environments have already saturated in performance and it may be interesting to try more complex planning tasks such as RLBench or other robotic manipulation environments.
>
> We agree that new benchmarks can provide more interesting insights. We chose our benchmarks based on
> (a) those used in related works like DD, Diffuser, TT,  and we further extended them to expose challenging variations (eg, POMDPs), and
> (b) the compute available in our academic setting.
>
> 1. We acknowledge that the D4RL benchmark is showing signs of saturation with many methods performing similarly. However, DS's top performance serves as a promising sanity check and indication of its relative strength.
>
> 2. To create more challenging scenarios, we developed POMDP environments that occlue two velocity dimensions. During the rebuttal, we conducted a detailed analysis to consider many other variations. The results, compared with all baselines in the POMDP section and presented in Table 1 of the rebuttal PDF, show that DS continues to outperform other baselines across different ways of dimension occlusions.
>
> 3. Lastly, our decision to use the Maze2D environment was done to evaluate the planning ability of DS. This is the key environment used for evaluating planning capabilities in our most closely related works, DD and Diffuser. The superior performance in this environment relative to baselines illustrates the strengths of our method in complex single and multi-goal planning tasks.
>
> We acknowledge the potential value of exploring other environments such as RLBench and will explicitly list it for future work in the final version.
>
> >Q2: “... I'm not sure I completely understand the particular decomposition of first observations, then rewards, and then actions. It seems like decomposing rewards first may be more natural then observations.”
>
> Thanks for the interesting comment. From the chain rule of probability, any autoregressive factorization can model the data distribution under idealized conditions. In practice, we choose the ordering of observations, rewards, and actions. Specifically, we ordered observations prior to rewards to be consistent with the functional definitions in MDPs, where the reward is typically a function of observations (and potentially other variables), but not vice versa. We tested this choice empirically as well in early experiments, and found our choice to significantly outperform the counterpart, which is shown in Table 3 in the rebuttal PDF. Moreover, while not the focus of our experiments, our choices allows us to directly use pretrained perception models for mapping goals to observations. These models are often trained with data that does not include reward information,  e.g., text2video models a language command (goal) to a sequence of video frames (observation sequence).
>
> >Q3: “...explore the extent to which decomposed submodules can enable compositional generalization or more efficient multitask learning by encoding structure in the greneration procedure. For instance -- perhaps the observation model can be used across a set of different tasks….”
>
> Thank you for this insightful suggestion! We included some evidence in this regard with the multi-goal experiments in Maze2D, where an agent trained only on single goal trajectory transfers to multi-goal environments in a zero-shot manner at test-time. In line with the reviewer’s advice, we conducted experiments during the rebuttal period for new forms of generalization of our decomposed modules to new environments, as we describe next.
>
> Specifically, we focused on the ability to transfer to other environments using shared observation models. We created modified versions of the Maze2D environment with varying action and reward spaces, while maintaining the same observation model. Our results demonstrate that DS facilitates an efficient learning process and compositional generalization. These findings further underline the modular efficiency of DS in scenarios where reusable components exist across different tasks or environments as shown in Table 2 in the rebuttal PDF.
>
> >Q4: In equation 6, we sequentially sample from each factored distribution to sample from equation 3. In practice, would we instead want to sample marginalized distribution over actions across all possible observations and rewards given the goal?
>
> We do not believe marginalization is necessary for planning-driven execution of the policy., For any planning algorithm, we are interested in sampling from $p(a_t, o_{t+1:T}, r_{t+1:T} | G, o_{0:t}, a_{0:t−1},r_{t+1:T})$, akin to randomized shooting approaches. In the case of Decision Stacks, ancestral sampling, as described in Eqs. 6-8. If we were interested in density estimation for $p(a_t | G, o_{0:t}, a_{0:t−1},r_{t+1:T})$, then we would need to marginalize over future observations and rewards.
>
> >Q5: "...how the individual decomposed modules can also provide feedback with respect to each other. For instance a planned set of states may induce a poor distribution over rewards. Having the conditional reward model then provide some feedback to regenerate the planned set of states may then be interesting."
>
> Thanks for the pointer and interesting idea! One potential execution strategy is to characterize poor reward distributions based on the extent to which they are out-of-distribution (ood)  from the training reward distribution. A highly ood sequence is likely to indicate poor planning of states, and we can exploit such a signal for replanning of states. While outside the scope of current work, we will include such a discussion in the paper for future investigation.

---

> > ### Comment · Reviewer_E8pR · 2023-08-18
> > **Thanks**
> >
> > Thanks for the clarifications! I think this paper is quite interesting and have raised my score to a 7.

---

> > > ### Author Response · Authors · 2023-08-18
> > > **Thank you**
> > >
> > > Thank you for acknowledging our work and the constructive and insightful feedback during the rebuttal discussions!

---

### Official Review · Reviewer_vCY4 · 2023-07-09

**Soundness:** 3 good
**Presentation:** 3 good
**Contribution:** 3 good
**Rating:** 6
**Confidence:** 3

**Summary:**

This paper proposes to disentangle the different modalities (reward, observations, and actions) utilized in offline goal-conditioned reinforcement learning instead of the standard temporal single-module structure. The paper contributes three independent generative modules that have the benefit of parallelizable training. The paper includes empirical results showing the proposed algorithm's comparable or superior performance to baselines on several MDP and POMDP environments.

**Strengths:**

1. The core idea to utilize a different decomposition of tokens to enable independent training of generative models appears interesting and novel. I enjoyed the simplicity of the approach, accompanied by the good empirical results.

2. Overall, I found the paper to be well-written and clear. I liked that the authors are clearly familiar with the relevant related literature and used diagrams throughout to help illustrate concepts.

3. Given the good empirical results, the idea of semantic decomposition (noted in strength 1) could be generally impactful for communities interested in transformers + RL.

4. I appreciate the extensive experiments performed to support the claims of flexibility and performance.

**Weaknesses:**

In general, I liked this paper. A few things that I think could improve it:

1. I would have liked to see the training cost comparison to the other algorithms.

2. In general, I would also like to see a more extensive discussion of the experimental results, as noted in questions 4-6 in the Questions section below.

3. I feel like some of the claims are a bit strong. For example, how does the work "guarantee" expressivity and flexibility? This language is often used when we have an assured property of the system or algorithm, but I don't see any rigorous theoretical evaluation of this claim.

More minor:

1. Missing some references, including work in compositional offline RL (as mentioned in the introduction, first paragraph) [Mendez et al., 2022] and a reference to the POMDP formalism [Kaelbling, Littman, and Cassandra, 1998].

2. The focus on MDP and POMDP environments was somewhat unclear to me. I think including some motivation or intuition in the Introduction as to why this approach would work better than previous works in both types of environments would make this more clear.

-----------------------------------------------------------------------------

[Mendez et al., 2022] Mendez, Jorge A., Harm van Seijen, and Eric Eaton. "Modular lifelong reinforcement learning via neural composition." arXiv preprint arXiv:2207.00429 (2022).

[Kaelbling, Littman, and Cassandra, 1998] Kaelbling, Leslie Pack, Michael L. Littman, and Anthony R. Cassandra. "Planning and acting in partially observable stochastic domains." Artificial intelligence 101.1-2 (1998): 99-134.

**Questions:**

1. Recent work [Carroll, et al. 2022] incorporates dynamics learning in some training setups and demonstrates improved performance over vanilla Decision Transformer. Is this not included in the experimental comparisons due to the different training scheme & architecture choice?

2. Why do only some of the algorithms have a reported error (e.g., in Tables 1&2)? Also, is the value standard error, standard deviation?

3. Could the authors please clarify what is meant by the expert-normalized scores in Section 4.2?

4. What is the training cost of the proposed algorithm (in terms of both samples and wall clock time)? How does it compare to the baselines?

5. Why is the architectural flexibility investigated using Hopper-medium-v2 despite it not being used in the other experiments (e.g., those presented in Table 3)?

6. Why does DS work better on medium but not medium expert or medium replay HalfCheetah (Table 3)?

7. How were the two dimensions of the state chosen to exclude (Section 4.3)? What is the observation representation in these experiments?

---------------------------------------------------------------------------------
[Carroll et al., 2022] Carroll, Micah, et al. "Uni [mask]: Unified inference in sequential decision problems." Advances in neural information processing systems 35 (2022): 35365-35378.

**Limitations:**

The authors have a small section in the Conclusion focusing on the limitations. I felt like this was sufficient. However, I can always appreciate a lengthier discussion section, especially when one's proposed algorithm performs similarly to other algorithms in some settings.

---

> ### Author Rebuttal · Authors · 2023-08-10
>
> Thank you for the valuable comments and very detailed insights! We address your questions below.
> >Q1: “...the training cost comparison to the other algorithms.”
>
> We provide a comparison in Table 4 of the rebuttal PDF, showcasing the per-iteration cost. DS's training time is influenced by the choice of generative models. Since the state model has a high dimensionality, and the modules can be trained in parallel, the training time for DS is determined by the state model's training time. DS's training cost aligns with those of DT and TT when using a transformer state model. It is the same as DD's and on par with Diffuser's training cost for a diffusion state model. This result affirms that DS offers an efficient approach without compromising on training cost.
>
> >Q2: “... some of the claims are a bit strong.., how does the work "guarantee" expressivity and flexibility? This language is often used when we have an assured property of the system or algorithm, but I don't see any rigorous theoretical evaluation of this claim.”
>
> We apologize for the apparent confusion here. For expressivity, we indeed have a theoretical guarantee that follows directly from our autoregressive factorization in Eq. 3. That is, since autoregressive models are derived from chain rule of probability, under idealized assumptions on model expressivity and dataset size, they can learn any data distribution. For flexibility, our claim is informal and refers to the property that the modular design allows us to use any architecture and training objective for the observation, reward, and action modules. We will edit and clarify the language in the final version.
>
> >Q3: “The focus on MDP and POMDP environments was unclear… include some motivation or intuition as to why this approach would work better than previous works in both types…”
>
> Both MDP and POMDP environments benefit from DS’s expressive modeling using the autoregressive factorization outlined in Eq. 3.
> For MDPs, we can apply the D-separation rule to the corresponding graphical model (a simplification of Fig 2 with no observation nodes) to recognize that the next action is not d-separated from the immediate reward when conditioned on the current observation and the task goal. This insight motivates us to model rewards in DS (unlike DD), which proves beneficial as shown in our ablation study in Table 5.
> For POMDPs, we can apply the D-separation rule to the corresponding graphical model (Fig 2) the D-separation rule to reveal that conditioning on the current observation and the goal does not obstruct the information pathway between the next action and prior or future observations, rewards, and actions, as demonstrated in Fig. 2. Thus, unlike prior works, the action and reward modules in DS by default condition on tokens from the past and future timesteps during training.
> Finally, DS's modular design respects the different modalities of states, actions, and rewards that typically span multiple dimensionalities and data types. This sets it apart from models like Diffuser, DT, and TT, which models them jointly with one architecture.
>
> >Q4: “Missing references [Mendez et al., 2022] and [Kaelbling, Littman, and Cassandra, 1998].”
>
> Thanks! We will add these referenced papers in an updated version of our paper.
>
> >Q5: Recent work [Carroll, et al. 2022] incorporates dynamics learning in some training setups and demonstrates improved performance over vanilla Decision Transformer. Is this not included in the experimental comparisons due to the different training scheme & architecture choice?
>
> Uni[MASK] falls into the transformer-based sequence modeling category, which we choose DT and TT for comparison. Uni[MASK] lacks reported results on locomotion tasks. Compared to Uni[MASK], DS also shows greater modularity. We see value in exploring comparisons with Uni[MASK] to deepen our understanding of training schemes.
>
> >Q6: Why do only some of the algorithms have a reported error (e.g., in Tables 1&2)? Also, is the value standard error, standard deviation?
>
> In Tables 1 and 2, values without reported errors are sourced from previous papers, while those with standard errors are from our experiments. We reported standard errors.
>
> >Q7: Could the authors please clarify what is meant by the expert-normalized scores in Section 4.2?
>
> The term "expert-normalized scores" in the D4RL benchmark suite refers to a normalization scheme applied to the performance scores, where the upper and lower bounds are with respect to a fixed "expert" policy and a fixed "random" policy.
>
> >Q8: Why is the architectural flexibility investigated using Hopper-medium-v2 despite it not being used in the other experiments (e.g…in Table 3)?
>
> We clarify that Hopper-medium-v2 was indeed used in the MDP and POMDP experiments, as in Tables 2 and 3. Choosing Hopper-medium-v2 for ablating architectural flexibility was arbitrary. The architectural flexibility experiment yields a 2x3x3 table of results where each entry is averaged over many random seeds, conducting on more environments would be computationally expensive.
>
> >Q9: Why does DS work better on medium but not medium expert or medium replay HalfCheetah (Table 3)?
>
> In Table 3, DS and other methods exhibit a performance trend on Halfcheetah dataset, with the order: medium expert > medium > medium-replay. This suggests a hierarchy in data quality, where medium-replay contains more suboptimal trajectories than medium, and medium has more than medium-expert.
> >Q10: “How were the two dimensions of the state chosen to exclude (Section 4.3)? What is the observation representation…?”
>
> Two velocity dimensions were excluded from the full state representation(consisting of positions and velocities) for POMDPs, simulating a lack of relevant sensors. A new sensitivity analysis on dimension occlusions further strengthens our results, as shown in Table 1 of the rebuttal PDF. DS continues to outperform other baselines.

---

### Official Review · Reviewer_Xfrm · 2023-07-28

**Soundness:** 3 good
**Presentation:** 4 excellent
**Contribution:** 2 fair
**Rating:** 5
**Confidence:** 3

**Summary:**

The paper proposes using different generative models (transformer or diffusion) rather than the same model (like in trajectory transformer/decision diffuser) for observation prediction, reward estimation, and action prediction in “model-based” offline RL. They demonstrate that this flexibility improves performance in offline RL in a POMDP setting where two dimensions of the state vector are removed for each environment.

**Strengths:**

1. The paper is well written and easy to read.
2. The consistent increase on POMDPs is encouraging. And the ablation with different modules is very interesting.

**Weaknesses:**

1. Similar performance to DD on D4RL Gym tasks.

2. Major similarities to previous work (trajectory transformer, decision diffuser) and the authors acknowledge this and highlight the reward modeling as novel. While it is interesting, a (temporal difference) variant already exists in the trajectory transformer paper [Janner et al 2021].  Namely, the trajectory transformer with Q function (a form of reward modeling) guided planning which can be found in Section 4.2 of [Janner et al 2021], under “Combining with Q functions”.

3. Does it matter which two observation dimensions are removed? An explanation of which dimensions were removed and why is missing and a sensitivity analysis to the dimensions removed are missing as well.

4. Is there any intuition for why the conditioning in Equations (4), (5), (6) were chosen?

5. Is there heuristic that could be used to choose the different modules with any online interaction? Such a heuristic would be necessary in offline RL.

**Questions:**

Please see weaknesses.

**Limitations:**

Yes.

---

> ### Author Rebuttal · Authors · 2023-08-10
>
> Thank you for your valuable comments and very detailed insights! We address your questions below.
> >Q1: Similar performance to DD on D4RL Gym tasks.
>
> 1. This similarity, rather than reflecting a limitation of our method, highlights the near-saturated state of the D4RL benchmark, where many methods perform similarly. However, DS's top performance serves as a promising sanity check of its strength, even if the gains are close.
>
> 2. Second, DS significantly surpasses other baseline methods in the POMDP setting, achieving a 76.9% improvement over DD and a 15.7% improvement over the next best method. During the rebuttal, a sensitivity analysis was conducted to consider various dimensions as detailed in Table 1 of the rebuttal PDF, showing that DS continues to outperform other baselines across different occlusions.
>
> 3. In the Maze2D environment, the superior performance relative to baselines illustrates the strengths of DS's planning ability in complex single and multi-goal planning tasks.
>
> 4. Apart from the above benchmark performance, we ran new experiments to demonstrate the compositional generalizability of DS which enables efficient transfer to new experiments, as detailed in Table 2 of the rebuttal PDF.
>
> >Q2: “Major similarities to previous work (trajectory transformer, decision diffuser) and the authors acknowledge this and highlight the reward modeling as novel… a (temporal difference) variant already exists in the trajectory transformer paper... the trajectory transformer with Q function (a form of reward modeling) guided planning which can be found in Section 4.2…”.
>
> We respectfully disagree with both assertions:
>
> **Reward Modeling is Not the Only Novelty**: the main novelty of our work extends beyond reward modeling. This includes:
>
> 1. DS introduces a novel token-first ordering of the trajectory components, instead of conventional timestep-first arrangements. This takes into account the distinct semantic attributes of each component, respecting their unique characteristics, such as domain type (discrete or continuous), and modality. While DT, TT and Diffuser chained states, action (and reward) with one model. Using a single model to model them together requires adjusting for modality-specific encoders and decoders, and close monitoring of training dynamics which can be avoided with DS.
>
> 2. Modular Expressivity and Compositional Generalization: As in Eq. 3, each module is chained autoregressively with subsequent modules, leading to maximal expressivity under idealized conditions. The absence of parameter sharing across modules also enables hardware parallelization and easy transfer to new environments with shared modules.
>
> 3. New experiments were conducted to showcase the compositional generalization of DS brought by the decomposed modules, as in Table 2 in the rebuttal PDF, where the same observation and/or reward models can be reused for different environments in Maze2D.
>
> 4. Flexible Generative Parameterization: Our method allows the use of various deep generative models, catering to real-world environments where agents may face challenges such as executing discrete actions given continuous observations.
> Superior Empirical Performance: The mentioned features translate into empirical superiority, as demonstrated in our results.
>
> **The TD Variant of TT cannot be used as justification to discount the novelty of our reward modeling approach**. While the Q-function in TT adopts a forward modeling approach, this necessitates dynamic programming. In contrast, DS takes a purely autoregressive route, signifying a clean and modular design. Unlike TT, which models cumulative returns, DS models per-time-step rewards. This differentiation is not trivial and translates into DS’s better empirical performance than TT.
>
> >Q3: explanation of which dimensions were removed and why is missing and a sensitivity analysis to the dimensions removed are missing as well.
>
> Thanks for suggesting! We did a new sensitivity analysis on dimension occlusions, considering various scenarios and comparing them with all the baselines in the POMDP section. In the paper’s POMDPs experiment, two specific velocity dimensions of the joints were excluded from the full state representation for all the locomotion environments, simulating lacking relevant sensors. The results in Table 1 of the rebuttal PDF show that DS continues to outperform other baselines in aggregate performance.
>
> >Q4: any intuition for why the conditioning in Equations (4), (5), (6) were chosen?
>
> From the chain rule of probability, any autoregressive factorization can model the data distribution under idealized conditions. In practice, we choose the ordering of observations, rewards, and actions. Specifically, we ordered observations prior to rewards to be consistent with the functional definitions in MDPs, where the reward is typically a function of observations (and potentially other variables), but not vice versa. We tested this choice empirically as well in early experiments, and found our choice to significantly outperform the counterpart, which is shown in Table 3 in the rebuttal PDF. Moreover, while not the focus of our experiments, our choices allows us to directly use pretrained perception models for mapping goals to observations. These models are often trained with data that does not include reward information,  e.g., text2video models a language command (goal) to a sequence of video frames (observation sequence).
>
> >Q5: any heuristic that could be used to choose the different modules with any online interaction?
>
> In offline RL, datasets guide the choice of module architecture based on data type. For example, diffusion models may be chosen for continuous image-based states, while transformer models may be more suitable for less continuous actions or rewards. DS’s modular design reflects the modality and data types, which may not be appropriately addressed by employing a singular transformer or diffusion model for joint state-action modeling.

---

> > ### Comment · Reviewer_Xfrm · 2023-08-11
> > **Response to the authors**
> >
> > >  DS introduces a novel token-first ordering of the trajectory components, instead of conventional timestep-first arrangements. Using a single model to model them together requires adjusting for modality-specific encoders and decoders.
> >
> > Does this simply shift the responsibility from the encoder to the tokenizer?
> >
> > > We did a new sensitivity analysis on dimension occlusions.
> >
> > This is really helpful. But is there perhaps a reason for occluding two dimensions only and not more? What happens with more occluded dimensions?
> >
> > > We tested this choice empirically as well in early experiments, and found our choice to significantly outperform the counterpart, which is shown in Table 3 in the rebuttal PDF.
> >
> > This is helpful as well for future users of DS.
> >
> > > In offline RL, datasets guide the choice of module architecture based on data type.
> >
> > I apologize for the typo. I meant to ask "any heuristic that could be used to choose the different modules **without** any online interaction?". I'm wondering about this because it is already particularly difficult to choose hyperparameters for a single model without any online interaction. Most papers today seem to evaluate online (indirectly violating offline RL requirements). Does having multiple architectures complicate this further?

---

> > > ### Author Response · Authors · 2023-08-13
> > > **question responses and a broader sensitivity analysis.**
> > >
> > > Thank you for the prompt response and careful reading of our rebuttal! We appreciate the new insights from your questions and address them below.
> > >
> > > >...simply shift the responsibility from the encoder to the tokenizer?
> > >
> > > Please apologize for our confusion but if we understand the question correctly, the comment is suggesting that our modules are different only in the use of different tokenizations? We’d argue that our modules are not simply employing distinct tokenizers.
> > > 1. As highlighted in the paper, splitting different modules explicitly avoids parameter sharing and thus provides more flexibility in training architectures, objectives, and optimization algorithms (e.g., transformers vs diffusion models). Such differentiation becomes important when considering the different natures of modalities. For instance, while states for robotic control can be represented as smooth signals in the form of images, the corresponding actions, such as joint torques, tend to be high-frequency and less smooth.
> > > 2. Using distinct modules in DS enables clean and effective generalization to new environments with overlapping semantics. In our rebuttal pdf (Table 2), we showed that the observation and reward models in DS, can be effectively reused across diverse environments with different action spaces. Doing so with non-modular approaches will involve finetuning the entire model.
> > > 3. Another advantage of our approach, as pointed out in our rebuttal, is that using distinct modules can allow us to directly use pretrained perception models. E.g., text2video models a language command (goal) to consecutive video frames (observation sequence).
> > >
> > > >What happens with more occluded dimensions?
> > >
> > > Thank you for pointing out the need for a broader sensitivity analysis! We conducted additional experiments and included the results below. We found that DS still continues to perform the best in aggregate performance.
> > > | Occluded dimension                     | BC       | DT        | TT        | DD       | Diffuser | DS(ours)  |
> > > |---------------------------------------|----------|---------- |---------- |----------|----------|---------- |
> > > | Occlude first 2 dim of velocity       | 43.3±2.2 | 59.0±2.3  | **68.7±8.3**  | 52.3±5.5 | 52.5±7.4 | 68.0±3.4 |
> > > | Occlude middle 2 dim of velocity      | 34.0±0.9 | 73.1±4.6  | 76.7±7.9  | 31.6±5.1 | 24.2±3.1 | **81.1±6.9** |
> > > | Occlude last 2 dim of velocity        | 51.1±1.3 | 105.2±1.0| 61.6±3.3 | 32.7±2.4 | 70.7±4.7 | **110.9±0.4**|
> > > | Occlude last 4 dim of velocity        | 2.5±0.1  | 53.6±4.8  | 59.3±6.8  | 9.0±2.6  | 5.8±1.7  | **69.0±9.9** |
> > > | Occlude last 6 dim of velocity        | 3.4±0.1  | 30.4±4.9  | **57.9±5.2**  | 3.8±8.1  | 2.9±0.2  | 33.0±4.7|
> > > | Occlude first 2 dim of position       | 14.6±2.4 | **40.4±3.2**  | 15.0±4.1  | 8.7±9.3  | 29.5±1.7 | 38.9±6.2 |
> > > | Occlude first 5 all dim of position   | 10.4±1.0 | 27.6±5.3  | 38.9±4.1  | 10.7±0.8 | 12.4±3.7 | **39.0±2.5** |
> > > | Average                               | 22.7±1.1 | 55.6±3.7  | 54.0±5.7  | 21.3±4.8 | 28.3±3.2 | **62.8±4.9** |
> > >
> > >
> > > > any heuristic that could be used to choose the different modules without any online interaction?
> > >
> > > Thank you for bringing up this important practical question! In addressing the challenge of choosing different modules in an offline RL setting without online interaction, we propose the following heuristics:
> > > 1. One potential heuristic for fine-tuning each module is to measure the module's validation likelihoods. These can be computed on a held-out set of offline trajectories, and crucially, they don't require online interactions. This approach facilitates a data-driven method to compare and select module architectures.
> > > 2. Similarly, we can use a 2-sample test (e.g., Wasserstein distance) to compare the generated sequences with the dataset sequences. Given DS's modular design, we believe we can effectively utilize traditional unimodal definitions for these tests. There's no need to make adjustments to accommodate multiple modalities, streamlining the evaluation process.
> > > 3. DS's modular design enables us to utilize models trained on similar data and modalities for design decisions. As each module is focused on a single modality, it's feasible to apply configurations from models previously pretrained on similar modalities. This can serve as a foundation, which can then be fine-tuned further based on specific project requirements.
> > >
> > > While we acknowledge that these heuristics may not guarantee an optimal performance correlation, they can significantly narrow down the search space.
> > >
> > > Finally, the modularity of DS eliminates some hyperparameters that are often encountered in non-modular designs. For instance, traditional designs may require optimization of a weighted average of per-variable loss terms, where the weights become additional hyperparameters to tune. For DS, each component can be individually inspected and optimized, as opposed to a non-modular system where issues might be more intertwined and challenging to pinpoint.

---

> > > > ### Author Response · Authors · 2023-08-18
> > > > **Reminder**
> > > >
> > > > Thank you again for your valuable feedback on our paper. We've carefully considered your comments and conducted new experiments to address the new questions you highlighted. We'd like to note that the discussion phase is drawing to a close, and we hope you've had a chance to review our response. Given the limited time, are there any other questions or concerns you'd like us to address? We would greatly appreciate your support in this regard.

---

> > > > > ### Comment · Reviewer_Xfrm · 2023-08-18
> > > > > **Thank you for the new experiments**
> > > > >
> > > > > Thank you for your response and new results. I am happy to raise my score from 3 to 5.
> > > > >
> > > > > With respect to my point about tokenizers, I simply meant to state that a token-first ordering of the trajectory components as proposed by DS may not necessarily be an improvement over the conventional timestep-first arrangement since now you require modality-specific tokenizers (such as BPE for language, patch tokenizer for images) rather than modality specific encoders (CNN for images, nn.Embedding for language).

---

> > > > > > ### Author Response · Authors · 2023-08-18
> > > > > > **Thank you**
> > > > > >
> > > > > > Thank you for acknowledging our work and the constructive and insightful feedback during the rebuttal discussions!

---

### Author Rebuttal · Authors · 2023-08-10

We thank all the reviewers for providing insightful feedback and constructive suggestions, including recommendations for new ablation studies. In alignment with these insightful suggestions, we have conducted and provided 4 new experimental results that clarify Decision Stacks's novelty and address the concerns raised in the reviews.

The summary of these experiments is as follows:

1. Table 1: **Sensitivity Analysis on Occluded Dimensions for POMDPs**. In the Hopper environment, the full state encompasses 5 dimensions of position data and 6 dimensions of velocity data across different joints. The table outlines experiments involving various dimensions and semantics for occlusion. DS consistently exhibits superior or second-best performance compared to other baselines on the hopper-medium-expert-v2 dataset.

     Results: **DS continues to outperform other baselines across different ways of dimension occlusions in aggregate performance.**



2. Table 2: **Experiments on Compositional Generalization in Maze2D**. With the absence of parameter sharing, the DS modular design allows efficient compositional generalization. This experiment investigates the transfer to other environments using shared observation models with diverse action and reward spaces. In the Maze2D environment, both dense and sparse rewards are analyzed, along with three variations of force applied to a 2D ball. These results emphasize DS's modular efficiency across scenarios where reusable components exist across different tasks or environments.

   Results: **DS modular design allows efficient compositional generalization**

3. Table 3: **Performance Comparison on Modeling Ordering**. From the chain rule of probability, any autoregressive factorization can model the data distribution under idealized conditions. In practice, we choose the ordering of states, rewards, and actions. Specifically, we ordered states prior to rewards to be consistent with the functional definitions in MDPs, where the reward is typically a function of observations (and potentially other variables), but not vice versa. We tested this choice empirically and found our choice to significantly outperform the counterpart. This table compares two different orderings: Reward-State-Action (R, S, A) and State-Reward-Action (S, R, A). The findings reveal that the S, R, A ordering outperforms the R, S, A ordering in the halfcheetah-medium-replay-v2 environment, underscoring DS's consistency with functional definitions in MDPs.


   Results: **DS's choice of S, R, A ordering outperforms the R, S, A ordering**


4. Table 4: **Per-Iteration Training Cost for Different Algorithms**. We provide a comparison of the per-iteration cost. DS's training time is influenced by the choice of generative models. Since the state model has a high dimensionality, and the modules can be trained in parallel, the training time for DS is determined by the state model's training time. Specifically, DS's training cost aligns with those of DT and TT (transformer-based) and is consistent with DD's and on par with Diffuser's (diffusion-based) training cost. This result further affirms that DS delivers an efficient approach without compromising on training cost.

   Results: **DS's training cost aligns with other baselines.**

Please refer to the attached PDF for details of the experiments

---

### Decision · Program_Chairs · 2023-09-21

**Decision:**

Accept (poster)

**Comment:**

The reviews and discussion offered mixed reactions and support. However, the clarity of presentation and analysis of the approach is relatively well done, and the modular design complements existing approaches. We do encourage the authors to use the feedback to revise and strengthen this work in the coming weeks.